# mRNABench: A curated benchmark for mature mRNA property and function prediction

**Ruian (Ian) Shi** [†,1,2,3], **Taykhoom Dalal** [†,3], **Philip Fradkin** [†,1,2],
**Divya Koyyalagunta** [3], **Simran Chhabria** [3], **Andrew Jung** [4],
**Cyrus Tam** [3], **Defne Ceyhan** [3], **Jessica Lin** [3], **Kaitlin U. Laverty** [2,3],
**Ilyes Baali** [3], **Bo Wang** [1,2,5], **Quaid Morris** [3]

[1] Department of Computer Science, University of Toronto, [2] Vector Institute,
[3] Computational and Systems Biology Program, Sloan Kettering Institute,
[4] Department of Electrical and Computer Engineering, University of Toronto,
[5] Peter Munk Cardiac Center, University Health Network

## Abstract

Messenger RNA (mRNA) is central to gene expression, and its half-life, localization, and translation efficiency drive phenotypic diversity in eukaryotic cells. While supervised learning has been used to study the mRNA regulatory code, self-supervised foundation models support a wider range of transfer learning tasks. However, the dearth of standardized benchmarks limits efforts to pinpoint the strengths of various models. Here, we present mRNABench, a benchmarking suite for mature mRNA biology, focused on human transcripts, that evaluates the representational quality of mature mRNA embeddings from self-supervised nucleotide foundation models. We curate 11 datasets and 79 prediction tasks that broadly capture salient properties of mature mRNA, and assess the performance of 24 families of nucleotide foundation models for a total of 259k experiments. Using these experiments, we study parameter scaling, correlations between sequence compressibility and performance, and data-splitting strategies. We identify synergies between two self-supervised learning objectives, and pre-train a new Mamba-based model that achieves state-of-the-art performance using $\sim 700\times$ fewer parameters. mRNABench can be found at: https://github.com/morrislab/mRNABench.

## 1 Introduction

Nucleotide foundation models show promise as general purpose embedding models for RNA transcripts, offering rich representations useful for diverse RNA function prediction tasks. Despite an abundance of foundation models, and corresponding benchmarks for DNA (Grešová et al., 2023; Zhou et al., 2023; Marin et al., 2023; Patel et al., 2024) and non-coding RNA (ncRNA) (Ren et al., 2024; Runge et al., 2024), the modelling of messenger RNA remains underexplored.

Mature messenger RNA (mRNA) is created by splicing, which selectively retains exonic regions from a pre-mRNA, creating an mRNA splice isoform. Alternative splicing is a combinatorial process that, from a single genomic locus, can generate multiple splice isoforms depending on exon choice, each with distinct properties and functions (Figure 1). Up to 90% of human genes are alternatively spliced (Wang et al., 2008), leading to substantial diversity of gene function among cells. Splicing dysregulation is implicated in cancer (Bradley & Anczuków, 2023)

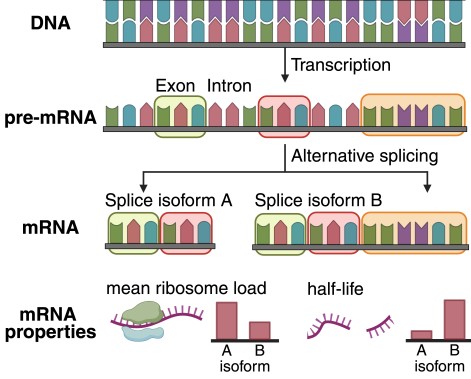

Figure 1: Genes are transcribed from DNA into pre-mRNA, which, through alternative splicing, can generate functionally distinct mRNA isoforms.

and other diseases (Chabot & Shkreta, 2016). More broadly, mRNA-based therapeutics such as mRNA vaccines (Sethna et al., 2025) are a rapidly growing area of drug development. Capturing complex aspects of mRNA biology through representation learning could accelerate scientific and therapeutic discovery, but meaningful progress requires benchmarking that reflects the unique features of mRNA.

Current DNA and ncRNA benchmarks are not well-suited for mRNA biology. The regulatory language encoded in mRNAs is distinct from that of DNA and ncRNA, so performance on DNA and ncRNA-centric benchmarks may not translate to mRNA-based tasks. Relevant mRNA functions are also distinct from those evaluated in ncRNA benchmarks, necessitating a novel collection of tasks that assess the representational ability of self-supervised mRNA foundation models.

To address this gap, we introduce MRNABENCH, a benchmarking suite with 11 distinct datasets containing 79 prediction tasks, designed to capture multiple facets of mRNA function and regulation. These tasks capture the most salient aspects of mRNA biology, including transcript stability, translational efficiency, localization, and post-transcriptional regulation. On these tasks, we use linear probing to evaluate 60 self-supervised nucleotide foundation models, covering the majority of publicly available models, and report insights in several key areas:

**Architectural Design:** We assess the impact of model size in nucleotide foundation models, and explore the suitability of contrastive learning and masked language modelling objectives for mRNA prediction tasks with differing contextual dependencies. Based on these findings, we add an MLM head to the Orthrus foundation model and optimally combine these objectives, achieving current state-of-the-art (SOTA) performance with $\sim 700\times$ fewer parameters than the best model.

**Sequence Compressibility:** We analyze the sequence content of mRNA and other genomic regions from the perspective of compression, underscoring the differences in regulatory grammar between regions. We correlate overall performance stratified by pre-training data source with these findings.

**Generalization:** As genomic benchmarking often suffers from homology or sequence similarity-based data leakage (Schreiber et al., 2020), we use biologically-aware data-splitting strategies to assess model generalization, and report performance overestimation.

In addition to the above insights, our overall contributions are:

- MRNABENCH, a Python package that provides lightweight access to 11 mRNA datasets with 79 function and property prediction tasks and code wrappers for 60 nucleotide foundation models. MRNABENCH provides integrated embedding and probing functionality, and is easily extensible to new datasets or models.
- Linear probe benchmarking of the above models on our curated datasets. We perform 259k experiments to assess the current state-of-the-art in mRNA modelling and analyze these results in terms of architecture choice, sequence compression, and generalization.

## 2 RELATED WORKS

**Deep learning for mRNA property prediction:** Supervised deep learning methods have been developed for prediction tasks such as mean ribosome loading (Sample et al., 2019; Karollus et al., 2021; Zheng et al., 2025a), half-life (Wayment-Steele et al., 2022; Agarwal & Kelley, 2022), subcellular localization (Wu et al., 2020; Garg et al., 2020), expression (Linder et al., 2025b), and RNA–protein interaction (Alipanahi et al., 2015; Horlacher et al., 2023), highlighting the diversity of relevant tasks in mRNA biology. Most commonly, these models use CNN-based architectures that are trained on labelled experimental data (Avsec et al., 2021; Pampari et al., 2024). While these models offer strong in-distribution prediction, they can overfit on technical noise or other dataset-specific signals. Supervised learning also typically has low sample-efficiency compared to transfer-learning approaches in mRNA property prediction (Fradkin et al., 2024), motivating self-supervised foundation models. We provide a detailed breakdown of all models evaluated in our benchmark in Appendix A.

**Benchmarks for Biological Sequences:** Several large-scale benchmarks now exist for DNA-based tasks (Grešová et al., 2023; Zhou et al., 2023; Marin et al., 2023; Patel et al., 2024), enabling comparisons across architectures, pre-training objectives, model sizes, etc. In contrast, few RNA-focused benchmarks have been introduced (Ren et al., 2024; Runge et al., 2024), and they primarily

evaluate RNA secondary structure and function prediction in non-coding RNAs (ncRNAs). Whereas ncRNA function often depends on secondary and tertiary structures (Sato et al., 2021), mRNAs are less structured (Rouskin et al., 2014) and are primarily regulated through linear sequence features (Glisovic et al., 2008; Mayr, 2017; Sasse et al., 2024). As a result, benchmarks that emphasize structural accuracy fail to capture core mRNA regulatory signals. Existing benchmarks (Ren et al., 2024) also focus on short sequences (hundreds of nucleotides), whereas mRNAs often span several kilobases, limiting evaluation of long-range dependencies and full-transcript representations.

Existing benchmarks can also fail to account for data leakage due to homology between genomic sequences. This can artificially inflate generalization performance, and even common strategies such as chromosomal hold-out have shown susceptibility to homology-based leakage (Rafi et al., 2025). In MRNABENCH, we explicitly quantify the extent of performance overestimation due to homology and evaluate how different splitting strategies affect estimates of model generalization.

## 3 BENCHMARKING TASKS

We curate 11 datasets, including 79 subtasks, summarized in Table 4. For each dataset, we use GenomeKit (DeepGenomics, 2025) to generate a six-track representation containing splice and codon positions. We further categorize our tasks as GLOBAL versus LOCAL based on how the signal giving rise to a label is distributed in the transcript across untranslated regions (UTRs) and coding sequences (CDS). GLOBAL tasks arise primarily from aggregate effects across multiple regulatory elements, whereas LOCAL tasks are dominated by local sequence features or perturbations (e.g., short motifs or single-nucleotide variants). While some tasks may span both regimes, this categorization provides a useful abstraction for studying how sequence information is integrated. We briefly describe each task below, and expand on data processing in Appendix B.

### 3.1 GLOBAL TASKS

**mRNA Half-Life (HL)** measures the time for half of the molecules of an mRNA transcript to degrade in the cell, and is a key determinant of transcript stability and gene expression. Longer-lived transcripts allow for more sustained protein production, whereas shorter-lived transcripts allow for rapid changes in gene expression in response to the cellular environment. We collect this data from (Agarwal & Kelley, 2022), which aggregates 66 human and mouse mRNA half-life experiments.

**Mean Ribosome Load (MRL)** represents the number of ribosomes associated with an mRNA transcript, serving as a proxy for its translational efficiency. Differential translation efficiency among transcripts enables post-transcriptional regulation, as two transcripts with identical expression levels can produce vastly different amounts of protein product. Our benchmark dataset is collected from a renal cell carcinoma cell line (RCC4/VHL) (Sugimoto & Ratcliffe, 2022).

**Paired mRNA Half-Life and Mean Ribosome Load (MRL-HL-Pair)** consists of synthetic mRNA sequences with paired measurements of mean ribosome load (MRL) and cellular half-life. These measurements were obtained using PERSIST-seq (Leppek et al., 2022), capturing sequence variation across the 5' UTR, CDS, and 3' UTR, and enable joint evaluation of two key mRNA properties. With only 203 samples, this task also serves as an assessment of model performance in low-data settings.

**Translation Efficiency (TE)** uses measurements from the RiboNN dataset (Zheng et al., 2025b), which quantifies translational output by normalizing ribosome density to mRNA abundance. TE values are derived from thousands of ribosome profiling and RNA-seq experiments, filtered for quality based on coding region coverage. Scores are aggregated across 78 human and 68 mouse cell types, and the mean TE is used for transcript-level evaluation.

**RNA Subcellular Localization (RNA-Loc)**: The subcellular compartment to which an mRNA localizes plays a crucial role in when and where its encoded protein is synthesized (Buxbaum et al., 2015). We include a dataset based on APEX-seq (Fazal et al., 2019), which uses an engineered enzyme to tag nearby RNAs at defined subcellular locations, enabling gene-level mapping across eight cellular compartments. We represent each gene using its APPRIS-defined principal transcript sequence (Rodriguez et al., 2013).

**RNA Lifecycle (RNA-Lifecycle)**: This task is derived from long-read, isoform-resolved direct RNA-sequencing data from (Ietswaart et al., 2024), which quantifies transcript flow through chromatin,

cytoplasm, and polysomes. We process these data to derive multilabel transcript-level targets indicating relative depletion in each compartment, defined as transcripts falling within the bottom 20% of flow-rate estimates. Depletion reflects regulatory modulation at different stages of the RNA lifecycle, including transcript processing, nuclear export, and translational initiation.

## 3.2 LOCAL TASKS

**Massively Parallel Translation Assay - Mean Ribosome Load (MRL-MPRA)** is based on an MPRA from (Sample et al., 2019), where synthetic 5' UTRs, either randomized or designed, were inserted upstream of reporter genes in human cells. The MRL was measured across multiple experimental conditions, including different RNA chemistries, library strategies, and reporter gene choices. Each subtask is framed as a regression predicting the measured mean ribosome load from sequence. In contrast to the previous MRL task, this task preserves the CDS while varying only the 5' UTR. Here, translation is largely driven by local features such as the presence of upstream AUGs.

**eCLIP Binding (eCLIP)** The eCLIP protocol (Van Nostrand et al., 2016) detects the binding positions of RNA-binding proteins (RBPs). RBPs regulate mRNA processing and function co- and post-transcriptionally, through alternative splicing (Tao et al., 2024), polyadenylation (Gruber et al., 2018), nuclear export (Soheilypour & Mofrad, 2016), stability (Li et al., 2022), among other processes. We process an eCLIP dataset using tracks collected from ENCODE (ENCODE, 2012) covering 168 RBPs across two cell lines. From this dataset, we identify the top 20 RBPs per cell line by number of events and simplify the task for linear probing by defining a binary classification task of whether each RBP binds to a given transcript.

**miRNA Binding (miRNA-Binding)** predicts whether an mRNA transcript is targeted by specific miRNAs. We use a dataset derived from miRTarCLASH (Yang et al., 2025), which captures miRNA–mRNA interactions by sequencing chimeric miRNA–target hybrids. To mitigate class imbalance, we restrict the benchmark to the 20 most prevalent miRNAs and frame the task as a set of per-miRNA binary classification problems. An mRNA transcript is included if it is targeted by at least one of these miRNAs.

**Variant Effect Prediction - TraitGym (VEP-TG)** evaluates detection of pathogenic single nucleotide variants (SNVs) within mature mRNA transcripts. We use a filtered subset of the TraitGym dataset (Benegas et al., 2025) restricted to UTR variants and retrieve the sequence of each sample using the APPRIS principal transcript (Rodriguez et al., 2013). Due to the restriction of sequence input to the mature mRNA region, the inherent predictability of SNV pathogenicity may be reduced.

**Variant Effect Prediction in UTR Regions (VEP-UTR)** uses a curated set of pathogenic and benign variants located in 5' and 3' UTRs from (Bohn et al., 2023). ClinVar variants were filtered by gnomAD allele frequency ($< 0.05$) and manually re-evaluated under ACMG guidelines to define the pathogenic set. The molecular mechanism for each variant was further examined to confirm that pathogenicity is mediated through UTR regions, and variants with confounding non-UTR effects were excluded.

## 4 METHODS

### 4.1 SELF-SUPERVISED LEARNING OBJECTIVES

Nucleotide foundation models aim to learn useful representations of biology using self-supervised learning (SSL) on unlabelled sequence data. In this work, we focus on masked language modelling (MLM) and contrastive learning (CL), later exploring their combination during pre-training.

**Masked language modeling** is an SSL objective in which a model is trained to reconstruct masked tokens based on the surrounding sequence context. Given an input sequence $x$ and a set of masked positions $\mathcal{M}$, the MLM loss is defined as:

$$\mathcal{L}_{\text{MLM}} = -\sum_{t \in \mathcal{M}} \log p(x_t \mid x_{\setminus \mathcal{M}}) \tag{1}$$

**Contrastive learning** learns global representations by bringing similar sequences (e.g. augmented views) closer in embedding space while pushing dissimilar ones apart. In this work, we use the Orthrus contrastive objective (Fradkin et al., 2024), which applies Decoupled Contrastive Learning (DCL) (Yeh et al., 2022) using splicing and orthology augmentations. Given two views of the same sample $z_i^1$ and $z_i^2$ (positive pair), unrelated samples $z_k$, and the temperature parameter $\tau$, the DCL objective is:

$$\mathcal{L}_{\text{DCL},i} = \log \left[ \sum_{k=1}^{N} \sum_{l=1}^{2} \mathbb{1}_{k \neq i} \exp \left( \frac{\langle z_i^1, z_k^l \rangle}{\tau} \right) \right] - w_i \frac{\langle z_i^1, z_i^2 \rangle}{\tau} \tag{2}$$

**Multi-task objectives:** In Section 5.1, we explore combining MLM and CL using a simple multi-task objective. While there are numerous strategies for multi-task optimization, recent work shows that simple scalarization (i.e. using fixed linear weights) is often the most effective in practice (Xin et al., 2022). For the Orthrus+MLM model later described in Section 5.1, we use the pre-training objective: $\mathcal{L} = \alpha \mathcal{L}_{\text{MLM}} + (1 - \alpha) \mathcal{L}_{\text{CL}}$. Here, $\alpha$ is a weight that balances the relative contribution of each objective, chosen based on the magnitude of the loss values at convergence in single-objective training to balance the contribution of each loss function.

## 4.2 DATA SPLITTING STRATEGIES

Random data splitting can overestimate model generalization due to data leakage from highly similar or homologous sequences (Rafi et al., 2025). For more rigorous assessment of generalization, MRNABENCH implements three biologically-informed data splitting strategies:

**Chromosomal hold-out** excludes sequences from training splits based on their chromosomal origin. While this remains a common strategy in genomics (Marin et al., 2023; de Almeida et al., 2025), it assumes functional independence between chromosomes. Recently, Rafi et al. (2025) showed that this can cause data leakage due to sequence redundancy and homology between chromosomes.

**k-mer-based splitting** clusters sequences with similar short subsequence patterns (k-mers). We compute k-mer frequency vectors and cluster sequences using k-means, assigning entire clusters to a single split. This reduces leakage of low-level sequence motifs important for tasks like RNA–binding prediction, where much of the prediction performance is driven by the presence of these k-mers.

**Homology-based splitting** groups genes that share a common evolutionary origin. Using paralogous gene pairs retrieved from the NCBI Gene database (Sayers et al., 2024), we apply a 35% sequence similarity threshold to filter low-confidence relationships and build transitive gene groups such that genes connected via paralogy are placed in the same data split. This is a stringent strategy to ensure that models are evaluated on unseen functional examples, though it can reduce training set diversity.

Each method has trade-offs: chromosomal holdout is the most commonly used but most prone to data leakage, k-mer splitting reduces motif leakage but may allow some sequence-level redundancy, and homology splitting provides the most stringent functional generalization test but assumes mRNA with similar functions will have similar properties, which may not always be true. In Table 1, we use homology splitting for all datasets except for the VEP, MRL-MPRA, and MRL-HL-Pair datasets, where we use naive random splitting instead. We evaluate the impact of each strategy in Section 5.3.

## 4.3 LINEAR PROBING

We use linear probing, i.e. training a linear classifier on frozen embeddings, to evaluate representation quality (Chen et al., 2020). Evaluating the linear separability of frozen embeddings measures how accessible pre-training information is in the learned features. While fine-tuning is a valid alternative, it entangles representation quality with downstream optimization choices and may result in poorer out-of-distribution generalization (Kumar et al., 2022). We compute transcript-level embeddings by averaging over per-nucleotide embeddings. For models with insufficient context length, input sequences are chunked. For VEP tasks, we computed embeddings for both wild-type and mutant sequences, and use the embedding difference as features. Our experimental setup is detailed in Appendix C.

## 5 RESULTS

We evaluate all foundation models, two naive baseline methods, and an *ab initio* supervised CNN baseline on all tasks using linear probing and report the mean performance across ten random data splits in Table 1. Prediction subtasks were mean-aggregated by their source dataset, and we use the best-performing model from each family. In further analysis, we report a model-specific overall performance by first applying a z-score transform to all model performances within each dataset, and then taking the mean across datasets. The Fisher transform was applied to Pearson correlations prior to standardization. Full results for all models, descriptions of baselines, chosen data splits, and standard errors are reported in Appendix A and D.

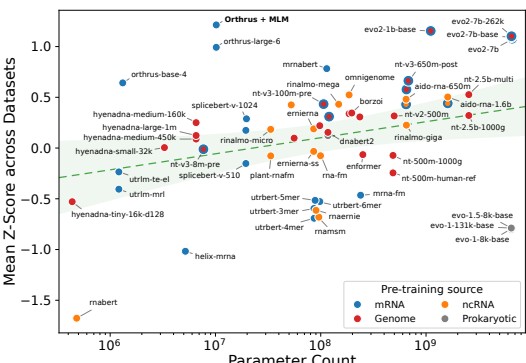

Figure 2: Linear probing performance. Mean of standardized metric across datasets shown on the y-axis. Data point colour shows pre-training source.

Table 1: Linear probe results. Mean of metric over ten random seeds reported. Best model per model family reported, see Appendix D for selected models. Best model for each dataset is highlighted and best foundation model is **underlined**. Models not significantly worse under Wilcoxon signed-rank test at $p = 0.05$ are **bolded**.

| | Local | | | | | Global | | | | | |
|---|---|---|---|---|---|---|---|---|---|---|---|
| | **VEP TG** AUPRC | **VEP UTR** AUPRC | **MRL MPRA** | **eCLIP** AUPRC | **miRNA Binding** AUPRC | **RNA Lifecycle** AUPRC | **RNA Loc** AUPRC | **HL** | **TE** | **MRL** | **MRL-HL-Pair** |
| Metric | (%) | (%) | R | (%) | (%) | (%) | (%) | R | R | R | R |
| Naive Baseline | 34.4 | 83.8 | 0.64 | 43.4 | 15.4 | 42.8 | 60.2 | 0.19 | 0.29 | 0.14 | 0.51 |
| Naive Mamba | 15.3 | 73.9 | 0.50 | 26.5 | 15.7 | 44.6 | 63.1 | 0.57 | 0.57 | 0.19 | 0.44 |
| Supervised CNN | 11.2 | 62.6 | **0.77** | 42.1 | 15.8 | 56.4 | 71.2 | 0.65 | 0.72 | 0.45 | 0.41 |
| 3UTRBERT | 17.4 | 72.8 | 0.60 | 31.1 | 15.3 | 44.5 | 63.3 | 0.40 | 0.55 | 0.19 | 0.50 |
| AIDO.RNA | 15.3 | 76.6 | 0.63 | 42.0 | 19.5 | 54.9 | 73.4 | 0.55 | 0.67 | 0.36 | **0.53** |
| Borzoi | 28.7 | 92.4 | 0.52 | 40.0 | 18.4 | 55.4 | 65.1 | 0.53 | 0.66 | 0.30 | 0.53 |
| DNABERT-S | 20.5 | 81.6 | 0.52 | 37.4 | 19.4 | 54.6 | 68.4 | 0.46 | 0.61 | 0.29 | 0.39 |
| DNABERT2 | 17.4 | 72.8 | 0.57 | 37.1 | 19.8 | 54.8 | 67.4 | 0.47 | 0.61 | 0.29 | 0.53 |
| ERNIE-RNA | 14.5 | 77.5 | 0.58 | 37.3 | 19.1 | 53.0 | 69.1 | 0.51 | 0.65 | 0.33 | 0.50 |
| Enformer | **29.6** | 81.4 | 0.46 | 37.1 | 17.3 | 51.3 | 63.4 | 0.46 | 0.62 | 0.26 | 0.44 |
| Evo1 | 19.6 | 83.8 | 0.27 | 26.2 | 16.1 | 45.0 | 63.4 | 0.33 | 0.47 | 0.20 | 0.33 |
| Evo2 | 26.8 | 86.7 | 0.67 | **46.1** | 19.0 | **60.7** | 77.0 | 0.63 | **0.75** | **0.44** | 0.57 |
| Helix-mRNA | 14.9 | 73.5 | 0.23 | 25.7 | 15.4 | 42.8 | 61.9 | 0.30 | 0.56 | 0.16 | 0.34 |
| HyenaDNA | 19.0 | 81.0 | 0.49 | 37.4 | **20.5** | 56.3 | 67.2 | 0.44 | 0.60 | 0.28 | **0.60** |
| mRNABERT | 26.2 | 82.2 | 0.69 | 40.1 | **20.2** | 57.4 | 71.9 | 0.59 | 0.70 | 0.38 | 0.46 |
| NT | 25.4 | 80.1 | 0.60 | 40.5 | 18.1 | 55.1 | 72.2 | 0.56 | 0.69 | 0.34 | **0.57** |
| NTv3 | 27.0 | 85.0 | 0.64 | 43.0 | 18.3 | 57.3 | 70.6 | 0.58 | 0.72 | 0.37 | 0.50 |
| OmniGenome | 17.3 | 61.8 | **0.78** | 39.0 | 19.1 | 57.1 | 70.7 | 0.53 | 0.65 | 0.36 | 0.54 |
| Orthrus | 17.3 | 76.5 | 0.63 | 43.6 | 19.8 | 60.0 | **77.9** | **0.67** | 0.72 | **0.43** | 0.51 |
| PlantRNA-FM | 15.9 | 75.5 | 0.62 | 33.6 | 17.9 | 50.7 | 66.4 | 0.45 | 0.58 | 0.25 | 0.46 |
| RNA-FM | 14.6 | 76.2 | 0.49 | 35.0 | 18.6 | 50.7 | 67.0 | 0.46 | 0.60 | 0.29 | 0.49 |
| RNA-MSM | 15.8 | 70.5 | 0.36 | 28.1 | 16.5 | 48.2 | 63.7 | 0.34 | 0.49 | 0.16 | 0.47 |
| RNABERT | 13.5 | 61.8 | 0.24 | 19.2 | 13.7 | 38.8 | 57.5 | 0.20 | 0.29 | 0.06 | 0.32 |
| RNAErnie | 16.8 | 76.3 | 0.33 | 29.0 | 16.7 | 48.3 | 63.7 | 0.33 | 0.47 | 0.14 | **0.55** |
| RiNALMo | 16.9 | 81.4 | 0.67 | 38.6 | 18.5 | 57.2 | 68.9 | 0.50 | 0.63 | 0.38 | 0.52 |
| SpliceBERT | 15.3 | 78.8 | 0.53 | 37.6 | 19.0 | 53.2 | 72.6 | 0.52 | 0.66 | 0.34 | **0.51** |
| UTR-LM | 14.4 | 75.8 | 0.54 | 33.7 | 18.1 | 51.3 | 65.8 | 0.43 | 0.57 | 0.25 | 0.38 |

We observe that Evo2 performs the best overall, with Orthrus, HyenaDNA, OmniGenome, and Borzoi outperforming on specific tasks. We explore Orthrus' weakness in local tasks in Section 5.1. In Figure 2, we visualize the mean standardized metric as a function of model parameters, and observe a weak correlation between model size and overall performance, although stratification reveals that other factors such as pre-training data source also contribute significantly to performance.

In contrast to results from the DNA foundation model space (Patel et al., 2024; Benegas et al., 2025), we find that the latest generation of RNA foundation models outperform simpler baseline and *ab initio* supervised methods in all but two of the tasks. While DNA language models must contend with learning from intergenic regions with low signal-to-noise ratios, the process of alternative splicing partially alleviates this issue, and consequently makes mRNA biology better suited for language-based approaches. However, we note that the regulatory language of mRNA still differs significantly from genomic DNA, and further explore this in Section 5.2.

## 5.1 JOINT PRE-TRAINING OBJECTIVES IMPROVE MRNA FOUNDATION MODELS

The competitive performance of the 10M-parameter Orthrus model relative to the 7B-parameter Evo2 model suggests that the choice of objective function plays a significant role in downstream performance. We observe that Orthrus mainly outperforms on global tasks, consistent with findings from computer vision that contrastive pre-training yields worse performance on finer-resolution tasks (Dippel et al., 2022; Cole et al., 2022). In Figure 3 (left), we quantify each model's global-task bias as the difference in mean z-score between global and local tasks. Orthrus shows a strong global-task bias, consistent with this property of contrastive learning.

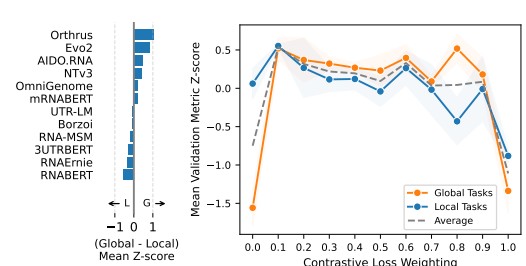

Figure 3: **Left:** Global task performance bias. **Right:** Performance of joint MLM-CL model as a function of contrastive objective weight.

To address this weakness in capturing local signals, we pre-train an mRNA foundation model that combines the Orthrus contrastive objective with an MLM loss. We investigate the optimal ratio between these two objectives using scalarization (Xin et al., 2022). To isolate the effect of the training objective, we fix the pre-training dataset and Mamba backbone from Orthrus and train models with varying weightings of the MLM and CL objectives. Experimental details are described in Appendix E.

Using the joint pre-training objective in Section 4.1, we trained models with CL-to-MLM ratios ranging from zero (MLM-only) to one (CL-only). As seen in Figure 3 (right), relying on a single objective, either MLM or CL in isolation, leads to poor overall results. Adding even a small amount of CL signal to the MLM-only model significantly improved global task performance. At low CL weights, the model's performance on local tasks was also unchanged, indicating that CL signal can be added to boost performance on global tasks without a corresponding drop in local task performance. Overall, we find that combining MLM and CL provides better task coverage than either alone.

Using validation scores, we select the best-performing joint model, denoted **Orthrus+MLM**, and find that it beats or matches the state-of-the-art foundation model in seven of 11 datasets (Table 2). Surprisingly, while the addition of the MLM head boosts overall performance, the largest gains are concentrated in global tasks, contrary to expectations, warranting further methodological exploration.

Table 2: Orthrus+MLM Results. Metrics reported identically to Table 1.

| | VEP TG | VEP UTR | MRL MPRA | eCLIP | miRNA Binding | RNA Lifecycle | RNA Loc | HL | TE | MRL | MRL-HL-Pair |
|---|---|---|---|---|---|---|---|---|---|---|---|
| Current SOTA FM | **29.6** | **92.4** | **0.78** | 46.1 | **20.5** | 60.7 | 77.9 | 0.67 | **0.75** | 0.44 | **0.60** |
| Orthrus+MLM | 18.1 | 75.7 | 0.64 | **46.5** | 20.0 | **62.8** | **78.9** | **0.69** | 0.75 | **0.46** | **0.63** |

## 5.2 REGIONAL SEQUENCE COMPRESSIBILITY CORRELATES WITH MODEL PERFORMANCE

Although nucleotide foundation models share the same input vocabulary, the sequence content and regulatory code of the genomic regions they represent differ substantially. Consequently, models trained on one region (e.g., genomic DNA) often perform poorly when applied to another (e.g., mRNA), as reflected in the aggregate model results when stratified by pre-training data source (Figure 4).

We quantify these distributional differences between regions using sequence compressibility. We apply Huffman coding to genomic sequences, using compressed code length to approximate entropy (Furht, 2006). Here, frequently occurring subsequences are assigned shorter code lengths, requiring fewer bits for representation. Prior work has shown that DNA and RNA are far less compressible than natural language, where higher compression ratios indicate stronger

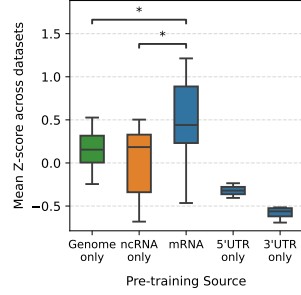

Figure 4: Performance stratified by pre-training source. Significance tested using Welch's t-test.

statistical regularities and underlying sequence structure (Schmitt & Herzel, 1997; Peng et al., 1992).

We constructed distinct Huffman coding schemes for various genomic regions: coding sequences (CDS), 5' untranslated regions (UTRs), 3' UTRs, non-coding RNA (ncRNA), intronic sequences, and intergenic DNA. 5' UTR sequences were the least compressible region, producing the lowest compression ratio (0.951), followed by CDS (0.953), consistent with the additional structure imposed by codon usage and tightly regulated translation initiation sequences (Parvathy et al., 2021; Hinnebusch et al., 2016). We evaluate the efficacy of each region-specific grammar in compressing data derived from the other genomic regions. A grammar obtained from the CDS compressed 3' UTR and intergenic sequences 10% and 14% less efficiently, respectively, indicating a significant distribution shift (Figure 5). In contrast, grammars originating from 3' UTR, intergenic, intronic, and ncRNA regions cross-compress with marginal loss in efficiency, suggesting similar sequence composition among these regions. Further analysis is described in Appendix F.

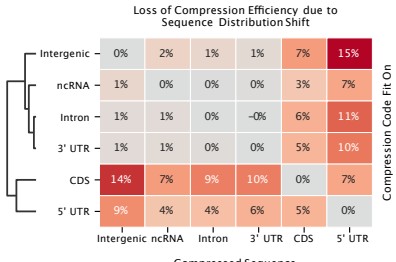

Figure 5: Cross-compression test scores by compression source. Values are percent increase in compression ratio.

These compression generalization gaps highlight the heterogeneity in sequence composition across different genomic regions. This analysis provides an empirical basis for understanding the generalization challenges observed when models pre-trained on non-coding sequences are applied to mRNA-specific tasks. Despite being components of mRNA, 3' UTRs exhibit a sequence composition resembling intergenic sequences rather than CDS and 5' UTR sequences. This distinction likely contributes to the model performance discrepancies observed in MRNABENCH.

## 5.3 BIOLOGICALLY INFORMED SPLITS IMPROVE GENERALIZATION ESTIMATES

One limitation in the evaluation of nucleotide foundation models lies in the overuse of naive random data splitting strategies, which tend to overestimate model generalization. This inflation arises because structurally or functionally related sequences often co-occur across data splits, inadvertently simplifying the predictive task. Given that functional outputs are frequently conserved among homologous or contextually similar sequences, models can achieve high performance by leveraging local sequence redundancy rather than by capturing broadly generalizable or mechanistically grounded features (Rafi et al., 2025).

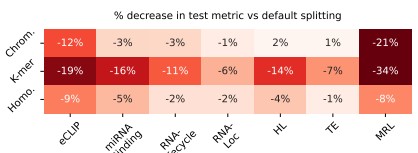

Figure 6: Average percent decrease in test metric across models and sub-tasks compared to naive random splitting.

To assess the impact of splitting strategy, we applied three biologically informed strategies across relevant tasks and compared them to random splitting. As shown in Figure 6, k-mer splitting consistently reduced performance, with homology-based splitting producing smaller but systematic generalization gaps. In some cases, trends aligned with biological expectations. For example, eCLIP and miRNA binding performance dropped more under k-mer splitting than homology splitting, which is consistent with our expectation (RBP/miRNA binding motifs are typically between 4 and 8 nucleotides long). These results indicate that naive random splits substantially overestimate generalization, and that biologically informed splits provide more realistic estimates of model robustness.

## 6 CONCLUSION

We present MRNABENCH, a benchmarking suite focused on probing mature mRNA function and regulation. Spanning 11 distinct datasets, 79 prediction tasks, and 259k experiments, it enables standardized evaluation across 60 nucleotide foundation models, leading to three key findings:

1. Larger models (e.g., Evo2) perform well, but Orthrus+MLM, a compact model with a biologically-informed joint objective, **matches or exceeds** their performance using over **700x fewer parameters**. This emphasizes the importance of design in addition to scaling.

2. Models trained on mRNA sequences consistently outperform those trained on DNA or ncRNA, highlighting the distributional differences between genomic regions. This is supported by compression-based analyses quantifying differences in sequence structure.

3. Naive evaluations (e.g., random splits) **overestimate generalization** and limit insight into what models have learned. Using biologically-aware splits, we show that current models fail to fully capture the modular structure of mRNA regulation.

Together, these results suggest a need to move beyond solely scaling, incorporating biological priors, and designing evaluations that probe for mechanistic - rather than correlative - understanding.

**Limitations and Future Work** Several limitations of our assessment exist. The assessment omits models that performed poorly after extensive troubleshooting (Li et al., 2023; Brixi et al., 2025) or had no accessible implementations (Saberi et al., 2024; Li et al., 2025). We envision future datasets on tasks such as 5' UTR structure (Rouskin et al., 2014; Loughrey et al., 2014) prediction will also further the completeness of our tasks. Due to the extensibility of MRNABENCH, we envision being able to add these datasets in future releases.

MEANINGFULNESS STATEMENT

A meaningful representation of life captures biologically relevant structure in a way that generalizes across contexts and supports mechanistic interpretation. For mRNA, this means encoding regulatory signals that govern stability, translation, localization, and other properties. This work contributes to this goal by providing a standardized benchmark that tests whether self-supervised nucleotide foundation models learn such signals across diverse, biologically grounded tasks and evaluation regimes. By exposing failure modes related to objective choice, pre-training data, and data leakage, MRNABENCH clarifies what current models do and do not represent about mature mRNA biology.

ACKNOWLEDGEMENTS

We are grateful to the High Performance Computing group at Memorial Sloan Kettering Cancer Center for providing assistance, support and compute resources; this work would have been impossible without them. Resources used in preparing this research were also provided, in part, by the Province of Ontario, the Government of Canada, through the Canadian Institute for Advanced Research (CIFAR) and companies sponsoring the Vector Institute.

This work is supported by an NIH Grant to Q.M. (R01 HG013328). This research was also funded in part through the NIH/NCI Cancer Center Support Grant P30 CA008748.

R.S., P.F., and A.J. are supported by a Vector Institute Research Grant. P.F. is supported by a Natural Sciences and Engineering Research Council of Canada (NSERC) Canada Graduate Scholarship (Doctoral Program). T.D. is supported by a National Science Foundation Graduate Research Fellowship under Grant No. 2139291. D.K is supported by a National Science Foundation Graduate Research Fellowship under Grant No. 227260-01. B.W. is supported by NSERC (grants: RGPIN-2020-06189 and DGECR-2020-00294), the Peter Munk Cardiac Centre AI Fund at the University Health Network and the CIFAR AI Chair Program.

We would like to acknowledge Stanley Z. Hua for preliminary exploration on eCLIP benchmarking.

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

APPENDIX CONTENTS

## A    OVERVIEW OF EVALUATED MODELS

All evaluated nucleotide foundation models are provided below. For each model family, written in bold, several variants are usually available, which tend to vary on model size or pre-training dataset. Where available, we used the `multimolecule` (Chen & Zhu, 2024) version of models.

| Model Name | Description | Pre-train Source | # Params (M) | Pre-training Context | Citation |
|---|---|---|---|---|---|
| **3'UTRBERT** | Transformer-based mRNA model pre-trained on 3'UTR regions of 100K human mRNA sequences. | - | - | - | Yang et al. (2024b) |
| utrbert-3mer | 3-mer tokenizer variant | mRNA | 86 | 512 | |
| utrbert-4mer | 4-mer tokenizer variant | mRNA | 87 | 512 | |
| utrbert-5mer | 5-mer tokenizer variant | mRNA | 88 | 512 | |
| utrbert-6mer | 6-mer tokenizer variant | mRNA | 98 | 512 | |
| **AIDO.RNA** | Transformer-based RNA model trained on 42M ncRNAs fron RNACentral. Some variants domain adapted for mRNA using 9M coding sequences. | - | - | - | Zou et al. (2024) |
| aido-rna-650m | - | ncRNA | 650 | 1024 | |
| aido-rna-1b600m | - | ncRNA | 1600 | 1024 | |
| aido-rna-650m-cds | CDS adapted variant | ncRNA, mRNA | 650 | 1024 | |
| aido-rna-1b600m-cds | CDS adapted variant | ncRNA, mRNA | 1600 | 1024 | |
| **Borzoi** | Hybrid Transformer and Convolutional, U-Net style DNA sequence to function model trained on human and mouse samples from ENCODE and GTEx data. The model was built to predict RNA-seq coverage from 524 kb genomic windows. | - | - | - | - |
| borzoi | - | Genome | 186 | 524,288 | Linder et al. (2025a) |
| flashzoi | Flashzoi is a variant of Borzoi trained with Flash attention. | Genome | 197 | 524,288 | Hingerl et al. (2025) |
| **DNABERT-S** | Transformer-based DNA model trained using contrastive learning on prokaryotic genomes. | Prokaryotic | 117 | N/A | Zhou et al. (2024) |
| **DNABERT2** | Transformer-based DNA model trained on multispecies dataset. Uses BPE and other modern architectural improvements for efficiency. | Genome | 117 | N/A | Zhou et al. (2023) |
| **ERNIE-RNA** | Transformer-based RNA model trained with structural information added as attention mask biases. Pretrained on 20M ncRNA sequences. | - | - | - | Yin et al. (2024) |
| ernie-rna | - | ncRNA | 86 | 1024 | |
| ernie-rna-ss | Fine-tuned on secondary structure. | ncRNA | 86 | 1024 | |
| **Enformer** | Hybrid Transformer and Convolutional DNA sequence to function model trained to predict over 5,000 human and 1600 mouse functional genomic tracks such as transcriptional activity, histone modifications, TF binding, and DNA accessibility | Genome | 251 | 196,608 | Avsec et al. (2021) |
| **Evo1** | StripedHyena-based DNA model autoregressively trained on 300b token OpenGenome dataset. | - | - | - | Nguyen et al. (2024) |
| evo-1-8k-base | Base model. | Prokaryotic | 6453 | 8,192 | |
| evo-1-131k-base | Trained with extended context. | Prokaryotic | 6453 | 131,072 | |
| evo-1.5-8k-base | Extends Evo1 dataset size by 50%. | Prokaryotic | 6453 | 8,192 | |

| Model Name | Description | Pre-train Source | # Params (M) | Pre-train Context | Citation |
|---|---|---|---|---|---|
| **Evo2** | StripedHyena2-based DNA model autoregressively trained on the OpenGenome2 dataset, containing 9T nucleotides across all domains of life. | - | - | - | Brixi et al. (2025) |
| evo2-1b-base | Base model trained on mixture of genomic regions including mRNA. | Genome, ncRNA, mRNA | 1000 | 8192 | |
| evo2-7b-base | Base model trained on mixture of genomic regions including mRNA. | Genome, ncRNA, mRNA | 7000 | 8,192 | |
| evo2-7b-262k | Intermediate context model trained on long genomic sequences. | Genome, ncRNA, mRNA | 7000 | 262,144 | |
| evo2-7b | Extended context model trained on long genomic sequences. | Genome, ncRNA, mRNA | 7000 | 1,000,000 | |
| evo2-40b | Despite passing author checks, we were unable to get good performance using either 40b models. | - | - | - | |
| **Helix-mRNA** | mRNA foundation model trained using a Mamba2 / Transformer hybrid backbone on 26M mRNAs from eukaryotic and viral species. | mRNA | 5 | 12,288 | Wood et al. (2025) |
| **HyenaDNA** | Hyena-based DNA model trained on human reference genome using an autoregressive scheme. Variants differ in size and context length. | - | - | - | Nguyen et al. (2023) |
| hyenadna-tiny-16k | - | Genome | 0.4 | 16,000 | |
| hyenadna-small-32k | - | Genome | 3 | 32,000 | |
| hyenadna-medium-160k | - | Genome | 7 | 160,000 | |
| hyenadna-medium-450k | - | Genome | 7 | 160,000 | |
| hyenadna-large-1m | - | Genome | 7 | 1,000,000 | |
| **mRNABERT** | Transformer-based RNA foundation model trained on 36M mRNA sequences using MLM (BERT-style) pre-training. It uses ALiBi positional embeddings and Flash Attention for efficient training. It was further pre-trained using contrastive learning to align the model's CDS embeddings with the corresponding protein embeddings from the ProtT5-XL-UniRef50 model. | mRNA | 114 | 1024 | Xiong et al. (2025) |
| **NucleotideTransformer (NT)** | Transformer-based DNA model with variants pre-trained on a differing datasets. Input tokenized to 6-mers. | - | - | - | Dalla-Torre et al. (2025) |
| 500m-human-ref | Trained on human reference. | Genome | 486 | 6,000 | |
| 500m-1000g | Trained on 1000 Genomes Project. | Genome | 486 | 6,000 | |
| 2.5b-1000g | Trained on 1000 Genomes Project. | Genome | 2550 | 6,000 | |
| 2.5b-multi-species | Trained on 850 genomes. | Genome | 2550 | 6,000 | |
| v2-50m-multi-species | v2 NT models use RoPE and GLUs. | Genome | 56 | 12,288 | |
| v2-100m-multi-species | - | Genome | 98 | 12,288 | |
| v2-250m-multi-species | - | Genome | 235 | 12,288 | |
| v2-500m-multi-species | - | Genome | 498 | 12,288 | |
| **NucleotideTransformerV3 (NTv3)** | Transformer/CNN-based, U-Net style DNA foundation model trained on the OpenGenome2 dataset using a masked language modeling objective at single nucleotide resolution. | - | - | - | Boshar et al. (2025) |
| v3-8M-pre | - | Genome, ncRNA, mRNA | 8 | 1,000,000 | |
| v3-100M-pre | - | Genome, ncRNA, mRNA | 100 | 1,000,000 | |
| v3-650M-pre | - | Genome, ncRNA, mRNA | 650 | 1,000,000 | |
| v3-100M-post | 100M parameter model further post-trained on 15,889 functional tracks for human, mouse, fruit fly, and six plant species covering chromatin state (i.e histone modifications) and transcriptional activity (i.e RNA-seq) | Genome, ncRNA, mRNA | 100 | 1,000,000 | |
| v3-650M-post | 650M parameter model further post-trained as above. | Genome, ncRNA, mRNA | 650 | 1,000,000 | |
| **OmniGenome** | Transformer-based RNA foundation model pretrained on plant mRNA sequences from the OneKP initiative, using single-nucleotide tokenization and ViennaRNA-predicted secondary structures. | - | - | - | Yang et al. (2024a) |
| omnigenome-52m | - | mRNA | 52 | 1024 | |
| omnigenome-186m | - | mRNA | 186 | 1024 | |

| Model Name | Description | Pre-train Source | # Params (M) | Pre-train Context | Citation |
|---|---|---|---|---|---|
| **Orthrus** | Mamba-based mRNA foundation model pre-trained on 40 mRNAs using contrastive learning on mRNA splice isoforms and orthologs. | - | - | - | Fradkin et al. (2024) |
| orthrus-base-4 | - | mRNA | 1 | 12,288 | |
| orthrus-large-6 | Pre-trained using splice and codon tracks. | mRNA | 10 | 12,288 | |
| **Plant-RNAFM** | Transformer-based RNA foundation model pretrained on 25M RNA sequences from 1,124 plant species (1KP). | mRNA | 33.5 | 1024 | Yu et al. (2024) |
| **RiNALMo** | Transformer-based ncRNA model trained on 36M ncRNAs. Architecture modernized to use RoPE, SwiGLU activations, and Flash Attention. | - | - | - | Penić et al. (2024) |
| rinalmo-micro | - | ncRNA | 33 | 1024 | |
| rinalmo-mega | - | ncRNA | 148 | 1024 | |
| rinalmo-giga | - | ncRNA | 650 | 1024 | |
| **RNABERT** | Transformer-based ncRNA model pre-trained using both MLM and a structural alignment learning objective on 80K ncRNA sequences. | ncRNA | 0.5 | 440 | Kalicki & Haritaoglu (2020) |
| **RNAErnie** | Transformer-based ncRNA model pre-trained on 23M ncRNA sequences. Uses motif-level masking during pre-training. | ncRNA | 86 | 512 | Wang et al. (2024) |
| **RNA-FM** | Transformer-based RNA foundation model pre-trained on 23M ncRNA sequences. Variant mRNA-FM trained only on CDS regions. | - | - | - | Chen et al. (2022) |
| rna-fm | - | ncRNA | 100 | 1024 | |
| mrna-fm | Trained on 45M CDS-only sequences | mRNA | 239 | 1024 | |
| **RNA-MSM** | Transformer-based ncRNA model trained using MSA from custom structure-based homology map on roughly 8M ncRNA sequences. | ncRNA | 86 | 1024 | Zhang et al. (2024) |
| **SpliceBERT** | Transformer-based mRNA model trained on 2M vertebrate mRNA sequences. Variants use only human RNA and using smaller contexts. | - | - | - | Chen et al. (2024) |
| splicebert-v-1024nt | - | mRNA | 20 | 1024 | |
| splicebert-v-510nt | - | mRNA | 19 | 510 | |
| splicebert-h-510nt | Trained only on human sequences. | mRNA | 19 | 510 | |
| **UTR-LM** | Transformer-based mRNA model that is pre-trained on random and endogenous 5'UTR sequences from various species. | - | - | - | Chu et al. (2024) |
| utrlm-te-el | Trained for translation efficiency and expression level. | mRNA | 1 | 1026 | |
| utrlm-mrl | Trained for mean ribosome loading. | mRNA | 1 | 1026 | |

Furthermore, we also implement two naive methods. NaiveBaseline uses manually extracted sequence features as embeddings. For the `naive-baseline-4` model, we take all k-mers from size three to seven as features, as well as the GC content of the sequence. The `naive-baseline-6` model additionally includes the CDS length and exon count as features. The NaiveMamba model is an Mamba with three layers and 64 hidden dimensions, without any pre-training (i.e., we use the "random" initial weights). We note that this works surprisingly well as an embedding model in comparison to the earliest generation of nucleotide foundation models.

Finally, we include an *ab initio* supervised dilated CNN comparison. This model is trained from scratch on the train split in evaluations. This supervised CNN uses a DilatedResNet architecture consiting of either three or four DilatedResNetBlocks. In each DilatedResNetBlock there are two DilatedConv1DBlocks. Each of these has two Conv1Ds layers with a kernel size of two and stride of two followed by BatchNorm1D layers, a dropout layer in-between, and residual connection followed by a MaxPool with kernel size of two. All layers are followed by ReLU activations. Each DilatedResNet block uses the same number of filters in convolutions, and has an associated dilation factor. We report these numbers for each dataset below.

For eCLIP, MRL-MPRA, HL, and MRL datasets, we use four DilatedResNetBlocks with the following hyperparameters:

```
dilation_params: [1, 2, 4, 8]
filter_nums: [64, 128, 128, 256]
```

For all other tasks, we use three DilatedResNetBlocks with the following hyperparameters:

```
dilation_params: [1, 2, 4]
filter_nums: [64, 128, 128]
```

Finally, all DilatedResNets have a stochastic shift layer which randomly shuffles nucleotides by up to three positions, and the final representations are projected using two layer 256 dimension projection feed forward network.

During training, we use the AdamW optimizer with a cosine warmup of 1000 steps. The chosen learning rate was 0.001, with a weight decay of 0.00001. Gradients were clipped was used at a threshold of five. We use an effective batch size of 1024, trained for 5k steps on MRL tasks, 500 steps for eCLIP, and 1.5k steps for all other tasks. These experiments were run on a mixture of A100, H100, and L40 GPUs, and each run took approximately 30 minutes to run.

## B   DATASET PROCESSING AND STATISTICS

### B.1   ECLIP

eCLIP data was downloaded from ENCODE (2012). In total, we downloaded 244 total eCLIP datasets, representing 168 unique targeted proteins; 139 of the experiments were in the K562 cell line, while the remaining 105 were in HepG2. For each eCLIP dataset, the most recent `BED` file was retrieved, and the corresponding `BAM` files from which the `BED` was derived were also retrieved. Given this data, we ran `Peakhood` (Uhl et al., 2022) on both the K562 and HepG2 RBP eCLIP datasets. We retrieved a genomic annotations file (`GTF`) from Ensembl (release 112) (Dyer et al., 2025) alongside an `hg38` genomic sequences file from UCSC (Karolchik et al., 2003), to input as arguments to the `-gtf` and `-gen` flags, respectively. The `-pre-merge` flag was also used. Of the 244 total eCLIP datasets, 223 of the experiments were from a paired-end eCLIP protocol, and thus were run using only R2 reads from the eCLIP data (`-bam-pp-mode 2`); the remaining 21 experiments were from a single-end eCLIP protocol and were run with no BAM preprocessing (`-bam-pp-mode 1`). Peakhood's output files include a set of files which identify which RNA-binding protein binding sites are associated with mature RNA transcript sequence binding (as opposed to pre-mRNA or intergenic binding sites). For this set of files, there are 2 groups, with the first group containing all possible transcript and binding site combinations and the second containing the most likely transcript sequence associated with a binding event. We chose this second group of transcript sequences as a high confidence set of sequences for the eCLIP binding task.

### B.2   MIRNA BINDING

The miRNA binding dataset is derived from MirTarCLASH (Yang et al., 2025), which experimentally captures miRNA-mRNA interactions by sequencing chimeric miRNA-target hybrids in human cells. We downloaded the full human MirTarCLASH interaction table and converted it into a transcript-level binary classification benchmark. To reduce class imbalance and focus on well-supported interactions, we restricted the benchmark to the 20 most frequently observed miRNAs in the dataset. For each miRNA, we define a binary target indicating whether a given mRNA transcript is experimentally validated as being targeted by that miRNA. Transcripts not targeted by any of the selected miRNAs were excluded. Transcript sequences were constructed using GENCODE v41 annotations via GenomeKit. For each gene, we retained up to five high-priority transcripts based on annotation confidence and selected a single representative transcript per gene to avoid over-representation. Transcripts lacking gene names or not present in the reference annotation were removed. The final benchmark consists of one mature mRNA transcript per gene, each annotated with binary targets corresponding to the presence or absence of binding for each of the 20 miRNAs. Sequence inputs include the full spliced mRNA sequence along with CDS and splice-site tracks.

### B.3    MEAN RIBOSOME LOAD - MPRA

This mean ribosome load (MRL) dataset is a massively parallel reporter assay (MPRA) that assesses the translational efficiency of random and designed 5' UTRs. The authors insert either randomized or designed 50-nucleotide sequences into a synthetic construct and measure MRL. We collected this data from the GEO repository for (Sample et al., 2019), which provides mean ribosome load measurements associated with synthetic transcripts varying in reporter gene, RNA chemistry, and 5' UTR sequence. Specifically, the eGFP and mCherry reporters are used; three RNA chemistries are evaluated (unmodified, pseudouridine, and m1-pseudouridine); and both random and designed 5' UTR sequences are included. We treat each unique combination of reporter gene, chemistry, and UTR design process as a separate sub-task, averaging MRL measurements across technical replicates. Sequences lacking measurements in both replicates were excluded.

### B.4    TRAIT GYM VARIANT EFFECT PREDICTION

Variant effect prediction (VEP) aims to classify single nucleotide polymorphisms (SNPs) as pathogenic or non-pathogenic. We use the TraitGym benchmark (Benegas et al., 2025) and subset it to only SNPs that occur in 5'UTR and 3'UTR regions. The TraitGym benchmark matches positive and negative samples based on sequence context (e.g., distance to the transcription start site). TraitGym also stratifies SNPs into Mendelian and complex traits. We group sequences according to this TraitGym stratification and treat them as two separate sub-tasks. Unlike the original dataset, we provide embedding models with a SNP's mature mRNA context rather than an unconstrained genomic context. To accomplish this, we searched the GENCODE v47 annotation using GenomeKit for mature mRNA transcripts that overlapped a specific SNP's locus and used the principal transcript according to APPRIS to generate the sequence context for the SNP. This may limit the overall predictability of this task, as SNPs may cause pathogenic effects prior to splicing that are not detectable in mature mRNA. We aim to filter out these SNPs in the future.

### B.5    5' AND 3' UTR VARIANT EFFECT PREDICTION

We use the dataset from Bohn et al. (2023) which consists of 26 3' UTR and 68 5' UTR variants with pathogenic or benign annotations confirmed via manual literature review. To calculate variant effect, we provide the models with the variant's mature mRNA context as in the TraitGym dataset above. We did this by retrieving the mature mRNA transcript sequence using the transcript ids provided by the authors from the NCBI RefSeq v109 annotation using GenomeKit.

### B.6    RNA HALF-LIFE

RNA half-life measures the time elapsed for half of the molecules of a given transcript to degrade. This dataset was collected from the Saluki paper (Agarwal & Kelley, 2022), which itself is an aggregate of 66 experimental datasets on human and mice. We treat each species as a separate sub-task. Due to the noisiness of RNA half-life measurements, the authors of Saluki instead propose regressing on the first principal component of the experiment-by-gene matrix. We collect the RNA half-life dataset from the Orthrus (Fradkin et al., 2024) data release on Zenodo: `https://zenodo.org/records/14708163`. We converted the six-track encoding back into a nucleotide sequence, and use GenomeKit to map transcripts to its chromosome using the gene name associated with each transcript. Sequences without a gene name were dropped. See the Orthrus paper for additional details on dataset composition.

### B.7    MEAN RIBOSOME LOAD

Mean ribosome load is a measure of the average number of ribosomes which are attached to a specific transcript, which is highly correlated with the translation rate of said transcript. We collect the MRL dataset from the Orthrus (Fradkin et al., 2024) data release on Zenodo: `https://zenodo.org/records/14708163`. The original data source for this dataset can be found at (Sugimoto & Ratcliffe, 2022). We converted the six-track encoding back into a nucleotide sequence, and use GenomeKit to map transcripts to its chromosome using the gene name associated with each transcript. See the Orthrus paper for additional details on dataset composition.

### B.8    TRANSLATION EFFICIENCY

Translation efficiency (TE) measures protein output per mRNA molecule and reflects regulation at the level of translation. We construct human and mouse translation efficiency benchmarks using transcript-level TE measurements from Zheng et al. (2025b). For each species, we matched transcript identifiers to reference annotations using GenomeKit and excluded transcripts without valid annotations or TE measurements. For each retained transcript, we generated the full mature mRNA sequence along with coding sequence (CDS) and splice-site tracks. Translation efficiency prediction is framed as a regression task using the provided TE values as targets.

### B.9    PAIRED MRNA HALF-LIFE AND MRL

We collect this data from (Leppek et al., 2022), which uses PERSIST-Seq to get paired measurements of mean ribosome load and RNA half-life for a specific synthetic sequence. Also reported were various predicted measures of mRNA secondary structure, which were unused in this study but could be a future prediction target. As the source data contained sequences for the 5'UTR, CDS, and 3'UTR regions in each synthetic transcript, we were able to reconstruct the full sequence and CDS tracks. The synthetic sequences did not undergo splicing. We drop all transcripts without both a HL and MRL measurement. During evaluation, we treat the prediction of MRL and HL as separate sub-tasks.

### B.10    RNA LIFECYCLE

The dataset used in this study (Ietswaart et al., 2024) investigates the dynamics of RNA as it is released from chromatin, exported from the nucleus, loaded onto polysomes, and degraded in both the nucleus and cytoplasm in human cells. FASTQ files from direct RNA sequencing (library prep kit SQK-RNA002) were downloaded from GEO accession `GSE208225`, comprising two replicates each from four categories: cytoplasm, nucleus, poly(A), and total RNA. For analysis, we used the `epi2me-labs/wf-transcriptomes` pipeline via Nextflow with the flags `-direct_rna True` and `-de_analysis`. Genomic annotations (GTF) were retrieved from GENCODE release 47, and the GRCh38 primary assembly genome sequence was used as the reference genome (`-ref_annotation` and `-ref_genome`, respectively). In the sample sheet, we designated `K562_cyto` and `K562_chr` as controls, and `K562_poly` and `K562_total` as treated samples to satisfy the pipeline requirements - we did not use any differential gene analysis output. Each sample was assigned a unique barcode as required for pipeline execution. The pipeline was run with default parameters. Output included an annotated assembled `transcriptome.fasta` file for each sample and a `transcripts_table.tsv` file containing isoform-level information along with differential gene and transcript analysis results. We only retained the transcripts table for each compartment for the purposes of the benchmark.

### B.11    RNA SUBCELLULAR LOCALIZATION

This dataset investigates the subcellular localization of mRNA transcripts using APEX-seq (Fazal et al., 2019). Each transcript is associated with proportions across eight cellular compartments: Nucleus, Nucleolus, Lamina, Nuclear Pore, Cytosol, ER membrane (ERM), Outer Mitochondrial Membrane (OMM), and ER Lumen. Although the original dataset was collected so that compartment labels were continuous proportions, we convert it into a multi-label classification task for simplicity in linear probing. Specifically, a compartment is considered positive if its associated proportion exceeds 0.125.

Table 4: Datasets included in MRNABENCH, grouped by task context. For benchmarking tasks with multiple subtasks, we report the range of dataset properties.

| Context | Benchmark | Task Type | # Unique Subtasks | # Unique Sequences | Mean Length | Max Length | Citation |
|---------|-----------|-----------|-------------------|--------------------|-------------|------------|----------|
| Local | eCLIP | Binary | 40 | 21559 | 3370 | 205012 | ENCODE (2012) |
| | miRNA-Binding | Binary | 20 | 12847 | 4188 | 101520 | Yang et al. (2025) |
| | MRL-MPRA | Regression | 6 | 710798 | 856 | 905 | Sample et al. (2019) |
| | VEP-TG | Classification | 2 | 4264 | 3659 | 28227 | Benegas et al. (2025) |
| | VEP-UTR | Classification | 2 | 234 | 4161 | 12070 | Bohn et al. (2023) |
| Global | HL | Regression | 2 | 26662 | 3549 | 12288 | Agarwal & Kelley (2022) |
| | MRL | Regression | 1 | 10,062 | 2620 | 12275 | Sugimoto & Ratcliffe (2022) |
| | TE | Regression | 2 | 21134 | 3723 | 17497 | Zheng et al. (2025b) |
| | MRL-HL-Pair | Regression | 2 | 203 | 991 | 1585 | Leppek et al. (2022) |
| | RNA-Lifecycle | Multilabel | 1 | 9957 | 2290 | 13459 | Ietswaart et al. (2024) |
| | RNA-Loc | Multilabel | 1 | 3335 | 4939 | 24319 | Fazal et al. (2019) |

## C  LINEAR PROBING EXPERIMENTAL SETUP

We conduct linear probes by first extracting sequence embeddings using the nucleotide foundation models described in Appendix A. Generally, models will provide a per-nucleotide or per-token embedding, and we compute the mean over the sequence length to get an $H$ dimensional vector per sequence. When model context length was insufficient to handle the full sequence, we computed embeddings by chunking sequences to the maximum possible length, and concatenating nucleotide embeddings across the sequence dimension prior to averaging. We found that strategies such as overlapping sequences between chunks has no major effect on performance.

Once embeddings were computed for all sequences in a dataset, we perform linear probing using Scikit-Learn's RidgeCV for regression tasks (CV performed on train and validation splits, over values $\alpha = $ 1e-3, 1e-2, 1e-1, 1, 10), LogisticRegression for classification tasks, and a MultiOutputClassifier with LogisticRegression for multi-label tasks. The evaluation metrics are micro-averaged for multi-label tasks. For tasks (MRL) and models (NaiveBaseline) where the design matrix is too large for normal ridge regression solvers, we use the `sag` solver.

We replicate this linear probing across ten different data splits, using seeds [2541, 2547, 413, 412, 411, 321, 421, 2515, 2516]. To aggregate performance over datasets, we first Z-transform metrics to account for differences in metric range in various tasks. Prior to standardizing Pearson correlations, we apply the Fisher transform. The Z-score is computed across models for a specific seed, dataset, and sub-task. We then mean-aggregate across sub-tasks and sub-datasets (e.g., grouping targets then cell lines for eCLIP). We then report the mean of these values across seeds. In Appendix D, we also report the 95% confidence interval using the standard error. For results that are reported by model group, we select the model with best overall performance in the group for visual clarity.

Runtime for linear probes were heavily dataset and model dependent. Individual linear probes can generally be run under one hour on a standard HPC CPU node, but the embedding process is highly variable. However, for most datasets and models, the dataset embeddings were also able to be generated within an hour on either an A100 or H100. We provide functionality within our code-base to embed sequences within a specific dataset in parallel for larger datasets.

# D    FULL LINEAR PROBING RESULTS

We report the results for each model variant, for each dataset group in Table 5 and 6. For full results, we provide a parquet file in the MRNABENCH GitHub repository with linear probing results for all split types, seeds, models, datasets, and tasks.

Table 5: Full linear probing results. Metrics are mean over ten random splits, 95% confidence intervals shown.

| Model | VEP TG AUPRC (%) | VEP UTR AUPRC (%) | MRL MPRA R | eCLIP AUPRC (%) | miRNA Binding AUPRC (%) |
|---|---|---|---|---|---|
| naive-4 | 0.344 ± 0.038 | 0.838 ± 0.073 | 0.642 ± 0.001 | 0.430 ± 0.004 | 0.158 ± 0.002 |
| naive-4-gc | 0.144 ± 0.023 | 0.743 ± 0.090 | 0.130 ± 0.001 | 0.176 ± 0.001 | 0.145 ± 0.002 |
| naive-4-kmer | 0.344 ± 0.038 | 0.838 ± 0.073 | 0.643 ± 0.001 | 0.430 ± 0.004 | 0.158 ± 0.002 |
| naive-6 | 0.344 ± 0.038 | 0.838 ± 0.073 | 0.642 ± 0.001 | 0.434 ± 0.004 | 0.154 ± 0.002 |
| naive-6-gc-struct | 0.144 ± 0.023 | 0.743 ± 0.090 | 0.130 ± 0.001 | 0.289 ± 0.004 | 0.171 ± 0.004 |
| naive-6-kmer | 0.344 ± 0.038 | 0.838 ± 0.073 | 0.643 ± 0.001 | 0.430 ± 0.004 | 0.158 ± 0.002 |
| naive-6-struct | 0.107 ± 0.016 | 0.539 ± 0.054 | NaN | 0.268 ± 0.004 | 0.161 ± 0.003 |
| naive-mamba | 0.153 ± 0.034 | 0.739 ± 0.087 | 0.499 ± 0.001 | 0.265 ± 0.002 | 0.157 ± 0.003 |
| dilated-resnet | 0.112 ± 0.006 | 0.626 ± 0.092 | 0.771 ± 0.008 | 0.421 ± 0.005 | 0.159 ± 0.003 |
| utrbert-3mer | 0.145 ± 0.022 | 0.827 ± 0.060 | 0.556 ± 0.001 | 0.296 ± 0.002 | 0.150 ± 0.002 |
| utrbert-4mer | 0.187 ± 0.021 | 0.776 ± 0.073 | 0.560 ± 0.002 | 0.286 ± 0.001 | 0.149 ± 0.002 |
| utrbert-5mer | 0.175 ± 0.029 | 0.720 ± 0.097 | 0.572 ± 0.001 | 0.313 ± 0.002 | 0.157 ± 0.003 |
| utrbert-6mer | 0.174 ± 0.029 | 0.728 ± 0.090 | 0.595 ± 0.002 | 0.311 ± 0.002 | 0.153 ± 0.003 |
| aido-rna-1b600m | 0.153 ± 0.033 | 0.766 ± 0.083 | 0.633 ± 0.001 | 0.420 ± 0.002 | 0.195 ± 0.003 |
| aido-rna-1b600m-cds | 0.146 ± 0.025 | 0.745 ± 0.080 | 0.645 ± 0.001 | 0.411 ± 0.003 | 0.188 ± 0.004 |
| aido-rna-650m | 0.179 ± 0.035 | 0.773 ± 0.085 | 0.645 ± 0.001 | 0.408 ± 0.002 | 0.189 ± 0.004 |
| aido-rna-650m-cds | 0.151 ± 0.027 | 0.701 ± 0.102 | 0.654 ± 0.001 | 0.412 ± 0.002 | 0.191 ± 0.004 |
| borzoi | 0.272 ± 0.045 | 0.885 ± 0.030 | 0.519 ± 0.002 | 0.405 ± 0.004 | 0.192 ± 0.003 |
| flashzoi | 0.287 ± 0.044 | 0.924 ± 0.038 | 0.515 ± 0.002 | 0.400 ± 0.002 | 0.184 ± 0.004 |
| dnabert-s | 0.205 ± 0.041 | 0.816 ± 0.046 | 0.516 ± 0.001 | 0.374 ± 0.002 | 0.193 ± 0.003 |
| dnabert2 | 0.174 ± 0.028 | 0.728 ± 0.072 | 0.570 ± 0.001 | 0.371 ± 0.002 | 0.198 ± 0.003 |
| ernierna | 0.145 ± 0.023 | 0.775 ± 0.078 | 0.580 ± 0.002 | 0.373 ± 0.002 | 0.191 ± 0.004 |
| ernierna-ss | 0.162 ± 0.027 | 0.804 ± 0.048 | 0.506 ± 0.002 | 0.358 ± 0.003 | 0.184 ± 0.004 |
| enformer-official-rough | 0.296 ± 0.046 | 0.814 ± 0.049 | 0.455 ± 0.004 | 0.371 ± 0.003 | 0.173 ± 0.003 |
| evo-1-131k-base | 0.203 ± 0.029 | 0.746 ± 0.073 | 0.276 ± 0.002 | 0.267 ± 0.003 | 0.163 ± 0.003 |
| evo-1-8k-base | 0.196 ± 0.041 | 0.838 ± 0.059 | 0.274 ± 0.002 | 0.262 ± 0.003 | 0.161 ± 0.003 |
| evo-1.5-8k-base | 0.235 ± 0.037 | 0.812 ± 0.064 | 0.271 ± 0.002 | 0.267 ± 0.003 | 0.162 ± 0.003 |
| evo2-1b-base | 0.268 ± 0.032 | 0.867 ± 0.050 | 0.666 ± 0.001 | 0.461 ± 0.004 | 0.190 ± 0.004 |
| evo2-7b | 0.261 ± 0.034 | 0.853 ± 0.043 | 0.699 ± 0.001 | 0.477 ± 0.005 | 0.183 ± 0.004 |
| evo2-7b-262k | 0.275 ± 0.029 | 0.869 ± 0.054 | 0.704 ± 0.001 | 0.478 ± 0.004 | 0.183 ± 0.004 |
| evo2-7b-base | 0.257 ± 0.032 | 0.864 ± 0.064 | 0.705 ± 0.001 | 0.473 ± 0.004 | 0.181 ± 0.004 |
| helix-mrna | 0.149 ± 0.031 | 0.735 ± 0.091 | 0.233 ± 0.002 | 0.257 ± 0.002 | 0.154 ± 0.002 |
| hyenadna-large-1m | 0.159 ± 0.031 | 0.778 ± 0.091 | 0.501 ± 0.001 | 0.374 ± 0.002 | 0.200 ± 0.003 |
| hyenadna-medium-160k | 0.189 ± 0.031 | 0.810 ± 0.085 | 0.493 ± 0.001 | 0.374 ± 0.002 | 0.205 ± 0.003 |
| hyenadna-medium-450k | 0.147 ± 0.029 | 0.784 ± 0.091 | 0.480 ± 0.001 | 0.376 ± 0.003 | 0.201 ± 0.003 |
| hyenadna-small-32k | 0.197 ± 0.032 | 0.805 ± 0.077 | 0.469 ± 0.002 | 0.353 ± 0.003 | 0.190 ± 0.003 |
| hyenadna-tiny-16k-d128 | 0.180 ± 0.025 | 0.746 ± 0.097 | 0.424 ± 0.002 | 0.298 ± 0.003 | 0.173 ± 0.003 |
| mRNABERT | 0.263 ± 0.036 | 0.822 ± 0.070 | 0.692 ± 0.001 | 0.401 ± 0.003 | 0.202 ± 0.004 |
| nt-2.5b-1000g | 0.177 ± 0.030 | 0.772 ± 0.068 | 0.554 ± 0.001 | 0.402 ± 0.003 | 0.198 ± 0.003 |
| nt-2.5b-multi-species | 0.254 ± 0.034 | 0.801 ± 0.064 | 0.602 ± 0.002 | 0.405 ± 0.003 | 0.181 ± 0.003 |
| nt-500m-1000g | 0.175 ± 0.032 | 0.747 ± 0.083 | 0.446 ± 0.002 | 0.353 ± 0.003 | 0.185 ± 0.002 |
| nt-500m-human-ref | 0.209 ± 0.030 | 0.754 ± 0.086 | 0.463 ± 0.002 | 0.367 ± 0.004 | 0.177 ± 0.004 |
| nt-v2-100m-multi-species | 0.190 ± 0.033 | 0.765 ± 0.074 | 0.507 ± 0.001 | 0.383 ± 0.003 | 0.193 ± 0.003 |
| nt-v2-250m-multi-species | 0.189 ± 0.034 | 0.731 ± 0.093 | 0.577 ± 0.001 | 0.385 ± 0.004 | 0.191 ± 0.003 |
| nt-v2-500m-multi-species | 0.179 ± 0.039 | 0.757 ± 0.084 | 0.571 ± 0.001 | 0.391 ± 0.003 | 0.193 ± 0.003 |
| nt-v2-50m-multi-species | 0.160 ± 0.037 | 0.683 ± 0.072 | 0.510 ± 0.002 | 0.385 ± 0.004 | 0.197 ± 0.003 |
| nt-v3-100M-post | 0.311 ± 0.039 | 0.846 ± 0.046 | 0.492 ± 0.002 | 0.386 ± 0.003 | 0.178 ± 0.002 |
| nt-v3-100M-pre | 0.186 ± 0.046 | 0.781 ± 0.087 | 0.612 ± 0.001 | 0.398 ± 0.002 | 0.202 ± 0.004 |
| nt-v3-650M-post | 0.270 ± 0.040 | 0.850 ± 0.064 | 0.639 ± 0.001 | 0.430 ± 0.003 | 0.183 ± 0.004 |
| nt-v3-650M-pre | 0.155 ± 0.025 | 0.661 ± 0.072 | 0.690 ± 0.001 | 0.417 ± 0.002 | 0.201 ± 0.004 |
| nt-v3-8M-pre | 0.135 ± 0.028 | 0.668 ± 0.073 | 0.573 ± 0.001 | 0.358 ± 0.002 | 0.190 ± 0.004 |
| omnigenome-186m | 0.173 ± 0.025 | 0.618 ± 0.053 | 0.775 ± 0.001 | 0.390 ± 0.002 | 0.191 ± 0.003 |
| omnigenome-52m | 0.158 ± 0.039 | 0.693 ± 0.063 | 0.759 ± 0.001 | 0.379 ± 0.003 | 0.187 ± 0.003 |
| orthrus-base-4 | 0.164 ± 0.030 | 0.822 ± 0.091 | 0.617 ± 0.001 | 0.413 ± 0.003 | 0.194 ± 0.002 |
| orthrus-large-6 | 0.173 ± 0.032 | 0.765 ± 0.060 | 0.629 ± 0.001 | 0.436 ± 0.004 | 0.198 ± 0.003 |
| orthrus+mlm | 0.181 ± 0.032 | 0.757 ± 0.092 | 0.639 ± 0.001 | 0.465 ± 0.003 | 0.200 ± 0.003 |
| plant-rnafm | 0.159 ± 0.036 | 0.755 ± 0.097 | 0.622 ± 0.001 | 0.335 ± 0.002 | 0.179 ± 0.003 |
| mrna-fm | 0.107 ± 0.016 | 0.539 ± 0.054 | 0.012 ± 0.003 | 0.375 ± 0.002 | 0.150 ± 0.003 |
| rna-fm | 0.146 ± 0.019 | 0.762 ± 0.087 | 0.487 ± 0.001 | 0.350 ± 0.002 | 0.186 ± 0.004 |
| rnamsm | 0.158 ± 0.032 | 0.705 ± 0.076 | 0.357 ± 0.002 | 0.281 ± 0.002 | 0.165 ± 0.003 |
| rnabert | 0.135 ± 0.017 | 0.618 ± 0.050 | 0.236 ± 0.002 | 0.192 ± 0.000 | 0.137 ± 0.002 |
| rnaernie | 0.168 ± 0.035 | 0.763 ± 0.089 | 0.325 ± 0.002 | 0.290 ± 0.002 | 0.167 ± 0.003 |
| rinalmo-giga | 0.229 ± 0.036 | 0.758 ± 0.087 | 0.742 ± 0.001 | 0.392 ± 0.003 | 0.158 ± 0.002 |
| rinalmo-mega | 0.169 ± 0.029 | 0.814 ± 0.081 | 0.673 ± 0.001 | 0.386 ± 0.003 | 0.185 ± 0.003 |
| rinalmo-micro | 0.177 ± 0.029 | 0.810 ± 0.070 | 0.600 ± 0.001 | 0.365 ± 0.002 | 0.184 ± 0.003 |
| splicebert-h-510nt | 0.174 ± 0.032 | 0.768 ± 0.054 | 0.567 ± 0.001 | 0.359 ± 0.002 | 0.184 ± 0.004 |
| splicebert-v-1024nt | 0.153 ± 0.032 | 0.788 ± 0.063 | 0.533 ± 0.002 | 0.377 ± 0.002 | 0.191 ± 0.004 |
| splicebert-v-510nt | 0.156 ± 0.034 | 0.792 ± 0.075 | 0.518 ± 0.002 | 0.331 ± 0.002 | 0.179 ± 0.004 |
| utrlm-mrl | 0.150 ± 0.034 | 0.743 ± 0.088 | 0.538 ± 0.001 | 0.308 ± 0.001 | 0.173 ± 0.003 |
| utrlm-te-el | 0.144 ± 0.041 | 0.758 ± 0.076 | 0.540 ± 0.002 | 0.337 ± 0.002 | 0.181 ± 0.004 |

Table 6: Full linear probing results. Metrics are mean over ten random splits, 95% confidence intervals shown.

| Model | RNA Lifecycle AUPRC (%) | RNA Loc AUPRC (%) | HL R | TE R | MRL R | MRL-HL-Pair R |
|---|---|---|---|---|---|---|
| naive-4 | 0.427 ± 0.007 | 0.596 ± 0.006 | 0.139 ± 0.012 | 0.268 ± 0.006 | 0.143 ± 0.007 | 0.510 ± 0.078 |
| naive-4-gc | 0.384 ± 0.007 | 0.606 ± 0.005 | 0.093 ± 0.008 | 0.149 ± 0.006 | -0.023 ± 0.015 | -0.046 ± 0.065 |
| naive-4-kmer | 0.427 ± 0.007 | 0.596 ± 0.007 | 0.139 ± 0.012 | 0.268 ± 0.006 | 0.143 ± 0.007 | 0.510 ± 0.078 |
| naive-6 | 0.428 ± 0.007 | 0.602 ± 0.006 | 0.195 ± 0.012 | 0.293 ± 0.007 | 0.145 ± 0.007 | 0.511 ± 0.073 |
| naive-6-gc-struct | 0.493 ± 0.007 | 0.614 ± 0.006 | 0.381 ± 0.008 | 0.480 ± 0.007 | 0.034 ± 0.010 | 0.140 ± 0.105 |
| naive-6-kmer | 0.427 ± 0.007 | 0.596 ± 0.007 | 0.139 ± 0.012 | 0.268 ± 0.006 | 0.143 ± 0.007 | 0.510 ± 0.078 |
| naive-6-struct | 0.471 ± 0.007 | 0.543 ± 0.008 | 0.362 ± 0.008 | 0.458 ± 0.008 | 0.038 ± 0.010 | 0.160 ± 0.092 |
| naive-mamba | 0.447 ± 0.009 | 0.631 ± 0.006 | 0.567 ± 0.006 | 0.566 ± 0.006 | 0.187 ± 0.017 | 0.437 ± 0.047 |
| dilated-resnet | 0.564 ± 0.012 | 0.712 ± 0.005 | 0.647 ± 0.004 | 0.715 ± 0.006 | 0.448 ± 0.021 | 0.411 ± 0.083 |
| utrbert-3mer | 0.443 ± 0.008 | 0.629 ± 0.008 | 0.395 ± 0.006 | 0.542 ± 0.008 | 0.157 ± 0.014 | 0.502 ± 0.056 |
| utrbert-4mer | 0.431 ± 0.006 | 0.623 ± 0.013 | 0.386 ± 0.009 | 0.533 ± 0.007 | 0.156 ± 0.016 | 0.430 ± 0.060 |
| utrbert-5mer | 0.461 ± 0.008 | 0.639 ± 0.009 | 0.393 ± 0.008 | 0.555 ± 0.008 | 0.171 ± 0.017 | 0.471 ± 0.053 |
| utrbert-6mer | 0.445 ± 0.006 | 0.633 ± 0.011 | 0.397 ± 0.008 | 0.549 ± 0.009 | 0.191 ± 0.015 | 0.497 ± 0.055 |
| aido-rna-1b600m | 0.549 ± 0.009 | 0.734 ± 0.008 | 0.546 ± 0.005 | 0.674 ± 0.006 | 0.360 ± 0.014 | 0.534 ± 0.074 |
| aido-rna-1b600m-cds | 0.538 ± 0.011 | 0.738 ± 0.009 | 0.559 ± 0.004 | 0.675 ± 0.004 | 0.378 ± 0.011 | 0.511 ± 0.044 |
| aido-rna-650m | 0.540 ± 0.007 | 0.730 ± 0.009 | 0.534 ± 0.005 | 0.669 ± 0.005 | 0.353 ± 0.013 | 0.503 ± 0.060 |
| aido-rna-650m-cds | 0.545 ± 0.007 | 0.740 ± 0.007 | 0.554 ± 0.006 | 0.677 ± 0.007 | 0.359 ± 0.015 | 0.489 ± 0.044 |
| borzoi | 0.556 ± 0.006 | 0.667 ± 0.006 | 0.497 ± 0.004 | 0.645 ± 0.004 | 0.313 ± 0.014 | 0.510 ± 0.039 |
| flashzoi | 0.554 ± 0.008 | 0.651 ± 0.008 | 0.527 ± 0.003 | 0.663 ± 0.004 | 0.296 ± 0.013 | 0.527 ± 0.051 |
| dnabert-s | 0.545 ± 0.007 | 0.685 ± 0.005 | 0.462 ± 0.006 | 0.610 ± 0.006 | 0.292 ± 0.015 | 0.393 ± 0.035 |
| dnabert2 | 0.548 ± 0.008 | 0.675 ± 0.008 | 0.470 ± 0.007 | 0.607 ± 0.005 | 0.286 ± 0.021 | 0.534 ± 0.050 |
| ernierna | 0.530 ± 0.010 | 0.691 ± 0.008 | 0.509 ± 0.004 | 0.646 ± 0.006 | 0.328 ± 0.017 | 0.503 ± 0.074 |
| ernierna-ss | 0.520 ± 0.008 | 0.670 ± 0.007 | 0.446 ± 0.008 | 0.581 ± 0.007 | 0.293 ± 0.014 | 0.532 ± 0.055 |
| enformer-official-rough | 0.513 ± 0.007 | 0.634 ± 0.009 | 0.463 ± 0.005 | 0.619 ± 0.005 | 0.263 ± 0.010 | 0.438 ± 0.054 |
| evo-1-131k-base | 0.457 ± 0.010 | 0.633 ± 0.007 | 0.328 ± 0.006 | 0.481 ± 0.006 | 0.185 ± 0.017 | 0.383 ± 0.061 |
| evo-1-8k-base | 0.450 ± 0.010 | 0.634 ± 0.008 | 0.327 ± 0.007 | 0.474 ± 0.006 | 0.198 ± 0.017 | 0.330 ± 0.049 |
| evo-1.5-8k-base | 0.453 ± 0.010 | 0.633 ± 0.008 | 0.315 ± 0.006 | 0.446 ± 0.007 | 0.193 ± 0.013 | 0.352 ± 0.073 |
| evo2-1b-base | 0.607 ± 0.009 | 0.770 ± 0.006 | 0.633 ± 0.006 | 0.752 ± 0.004 | 0.441 ± 0.013 | 0.572 ± 0.053 |
| evo2-7b | 0.591 ± 0.009 | 0.760 ± 0.007 | 0.647 ± 0.006 | 0.759 ± 0.002 | 0.447 ± 0.007 | 0.527 ± 0.077 |
| evo2-7b-262k | 0.593 ± 0.009 | 0.761 ± 0.007 | 0.646 ± 0.006 | 0.759 ± 0.003 | 0.444 ± 0.008 | 0.505 ± 0.083 |
| evo2-7b-base | 0.594 ± 0.008 | 0.767 ± 0.007 | 0.649 ± 0.006 | 0.758 ± 0.004 | 0.455 ± 0.009 | 0.507 ± 0.066 |
| helix-mrna | 0.428 ± 0.007 | 0.619 ± 0.006 | 0.297 ± 0.005 | 0.555 ± 0.004 | 0.155 ± 0.026 | 0.339 ± 0.047 |
| hyenadna-large-1m | 0.562 ± 0.008 | 0.670 ± 0.007 | 0.453 ± 0.008 | 0.604 ± 0.006 | 0.287 ± 0.014 | 0.506 ± 0.060 |
| hyenadna-medium-160k | 0.563 ± 0.008 | 0.672 ± 0.009 | 0.436 ± 0.006 | 0.599 ± 0.007 | 0.281 ± 0.013 | 0.603 ± 0.060 |
| hyenadna-medium-450k | 0.554 ± 0.008 | 0.674 ± 0.007 | 0.451 ± 0.005 | 0.611 ± 0.008 | 0.306 ± 0.012 | 0.460 ± 0.059 |
| hyenadna-small-32k | 0.521 ± 0.008 | 0.658 ± 0.008 | 0.418 ± 0.003 | 0.585 ± 0.007 | 0.218 ± 0.018 | 0.625 ± 0.037 |
| hyenadna-tiny-16k-d128 | 0.479 ± 0.010 | 0.639 ± 0.006 | 0.381 ± 0.004 | 0.540 ± 0.005 | 0.178 ± 0.015 | 0.440 ± 0.071 |
| mRNABERT | 0.575 ± 0.010 | 0.719 ± 0.010 | 0.586 ± 0.005 | 0.701 ± 0.005 | 0.382 ± 0.014 | 0.464 ± 0.039 |
| nt-2.5b-1000g | 0.560 ± 0.008 | 0.699 ± 0.008 | 0.488 ± 0.006 | 0.632 ± 0.005 | 0.313 ± 0.024 | 0.560 ± 0.051 |
| nt-2.5b-multi-species | 0.551 ± 0.010 | 0.722 ± 0.008 | 0.562 ± 0.004 | 0.687 ± 0.006 | 0.343 ± 0.012 | 0.567 ± 0.053 |
| nt-500m-1000g | 0.524 ± 0.006 | 0.656 ± 0.008 | 0.397 ± 0.004 | 0.554 ± 0.004 | 0.216 ± 0.019 | 0.511 ± 0.063 |
| nt-500m-human-ref | 0.520 ± 0.007 | 0.669 ± 0.009 | 0.446 ± 0.006 | 0.593 ± 0.006 | 0.249 ± 0.016 | 0.569 ± 0.055 |
| nt-v2-100m-multi-species | 0.547 ± 0.006 | 0.704 ± 0.007 | 0.484 ± 0.005 | 0.635 ± 0.006 | 0.269 ± 0.016 | 0.516 ± 0.054 |
| nt-v2-250m-multi-species | 0.555 ± 0.006 | 0.714 ± 0.009 | 0.509 ± 0.007 | 0.650 ± 0.005 | 0.308 ± 0.015 | 0.535 ± 0.047 |
| nt-v2-500m-multi-species | 0.553 ± 0.007 | 0.719 ± 0.008 | 0.525 ± 0.004 | 0.660 ± 0.003 | 0.297 ± 0.009 | 0.512 ± 0.051 |
| nt-v2-50m-multi-species | 0.551 ± 0.007 | 0.691 ± 0.008 | 0.471 ± 0.004 | 0.620 ± 0.006 | 0.269 ± 0.019 | 0.487 ± 0.047 |
| nt-v3-100M-post | 0.563 ± 0.006 | 0.684 ± 0.008 | 0.495 ± 0.006 | 0.646 ± 0.005 | 0.304 ± 0.015 | 0.435 ± 0.072 |
| nt-v3-100M-pre | 0.533 ± 0.009 | 0.721 ± 0.009 | 0.539 ± 0.006 | 0.677 ± 0.006 | 0.320 ± 0.016 | 0.487 ± 0.080 |
| nt-v3-650M-post | 0.573 ± 0.007 | 0.706 ± 0.010 | 0.581 ± 0.004 | 0.716 ± 0.004 | 0.369 ± 0.013 | 0.500 ± 0.059 |
| nt-v3-650M-pre | 0.558 ± 0.009 | 0.739 ± 0.007 | 0.587 ± 0.005 | 0.717 ± 0.006 | 0.352 ± 0.013 | 0.466 ± 0.069 |
| nt-v3-8M-pre | 0.507 ± 0.009 | 0.692 ± 0.011 | 0.510 ± 0.005 | 0.637 ± 0.005 | 0.285 ± 0.012 | 0.401 ± 0.063 |
| omnigenome-186m | 0.571 ± 0.007 | 0.707 ± 0.009 | 0.526 ± 0.004 | 0.647 ± 0.006 | 0.361 ± 0.017 | 0.537 ± 0.054 |
| omnigenome-52m | 0.565 ± 0.009 | 0.686 ± 0.007 | 0.502 ± 0.005 | 0.643 ± 0.004 | 0.355 ± 0.019 | 0.542 ± 0.059 |
| orthrus-base-4 | 0.583 ± 0.008 | 0.750 ± 0.007 | 0.537 ± 0.007 | 0.672 ± 0.006 | 0.363 ± 0.012 | 0.544 ± 0.061 |
| orthrus-large-6 | 0.600 ± 0.007 | 0.779 ± 0.006 | 0.666 ± 0.005 | 0.723 ± 0.006 | 0.427 ± 0.011 | 0.513 ± 0.089 |
| orthrus+mlm | 0.628 ± 0.008 | 0.789 ± 0.005 | 0.692 ± 0.004 | 0.750 ± 0.005 | 0.460 ± 0.012 | 0.633 ± 0.047 |
| plant-rnafm | 0.507 ± 0.010 | 0.664 ± 0.007 | 0.453 ± 0.004 | 0.583 ± 0.005 | 0.254 ± 0.014 | 0.456 ± 0.080 |
| mrna-fm | 0.530 ± 0.006 | 0.694 ± 0.007 | 0.535 ± 0.006 | 0.684 ± 0.005 | 0.321 ± 0.009 | 0.252 ± 0.068 |
| rna-fm | 0.507 ± 0.009 | 0.670 ± 0.006 | 0.462 ± 0.005 | 0.598 ± 0.008 | 0.292 ± 0.017 | 0.494 ± 0.071 |
| rnamsm | 0.482 ± 0.008 | 0.637 ± 0.006 | 0.340 ± 0.007 | 0.493 ± 0.006 | 0.157 ± 0.016 | 0.474 ± 0.042 |
| rnabert | 0.388 ± 0.007 | 0.575 ± 0.006 | 0.203 ± 0.009 | 0.288 ± 0.006 | 0.056 ± 0.013 | 0.322 ± 0.065 |
| rnaernie | 0.483 ± 0.009 | 0.637 ± 0.005 | 0.334 ± 0.007 | 0.474 ± 0.006 | 0.140 ± 0.016 | 0.550 ± 0.046 |
| rinalmo-giga | 0.485 ± 0.009 | 0.690 ± 0.009 | 0.525 ± 0.006 | 0.527 ± 0.006 | 0.416 ± 0.021 | 0.466 ± 0.066 |
| rinalmo-mega | 0.572 ± 0.009 | 0.689 ± 0.011 | 0.502 ± 0.005 | 0.630 ± 0.006 | 0.381 ± 0.018 | 0.518 ± 0.049 |
| rinalmo-micro | 0.559 ± 0.008 | 0.669 ± 0.007 | 0.438 ± 0.005 | 0.579 ± 0.006 | 0.348 ± 0.018 | 0.479 ± 0.048 |
| splicebert-h-510nt | 0.511 ± 0.010 | 0.720 ± 0.009 | 0.517 ± 0.005 | 0.644 ± 0.005 | 0.297 ± 0.015 | 0.512 ± 0.058 |
| splicebert-v-1024nt | 0.532 ± 0.010 | 0.726 ± 0.010 | 0.524 ± 0.003 | 0.656 ± 0.006 | 0.336 ± 0.013 | 0.514 ± 0.056 |
| splicebert-v-510nt | 0.477 ± 0.008 | 0.679 ± 0.007 | 0.451 ± 0.004 | 0.560 ± 0.008 | 0.238 ± 0.014 | 0.575 ± 0.059 |
| utrlm-mrl | 0.493 ± 0.010 | 0.648 ± 0.007 | 0.379 ± 0.005 | 0.534 ± 0.006 | 0.223 ± 0.015 | 0.412 ± 0.078 |
| utrlm-te-el | 0.513 ± 0.009 | 0.658 ± 0.007 | 0.433 ± 0.007 | 0.567 ± 0.006 | 0.249 ± 0.012 | 0.379 ± 0.070 |

# E  JOINT OBJECTIVE EXPERIMENTAL SETUP

We perform an ablation study on the mixture of contrastive learning (CL) and masked language modelling (MLM) losses using the Orthrus model as a controlled architecture. Here, we use the architecture from the orthrus-large-6 model, which is a Mamba model with six layers and a hidden dimension of 512. See the Orthrus paper for further optimizer hyperparameters, which we matched exactly. We use a subset of the Orthrus pre-training dataset, which is uses RefSeq annotations for ten

species to generate positive pairs for contrastive learning. In total, this results in approximately 1M mRNA sequences.

To ablate the balance of the MLM and CL objectives, we first trained models with only MLM and CL, and assessed the gradient magnitude upon convergence. We then selected values of $\alpha$ for the equation in Section 5.1 balancing MLM and CL such that the converged gradient magnitude for MLM and CL would be equal when $\alpha = 0.5$. We then interpolated $\alpha$ from 0 to 1 in increments of 0.1. For each value of $\alpha$, we pre-train three models using different seeds, for a total of 30 models. Each Orthrus pre-training run took approximately 16 hours using 4 H100 GPUS. We then select the best model for further evaluation using a simple weighted average of validation loss between global and local tasks, which was the $\alpha = 0.1$ model. Here, the validation loss was computed similarly to other overall metric aggregations, including the use of identical random data splitting seeds. Finally, we note that the shaded area shown in Figure 3 (right) shows standard error of these mean validation losses.

## F    GENOMIC DATA COMPRESSION

We define the Compression Ratio $CR$ for a given sequence $s$ using a specific compression model $M$. This model $M$ (e.g., a Huffman codebook) is typically derived from training on a representative set of sequences from a particular source distribution (e.g., CDS, 5' UTRs). The $CR$ quantifies the efficiency of the compression by comparing the total number of bits required to represent the sequence using model $M$ to its original size.

Let the sequence $s$ consist of $N = |s|$ tokens (e.g., k-mers), denoted as $s^1, s^2, ..., s^N$, $L_M = (s^k)$ be the code length (in bits) assigned to the kth token $s^k$ by the model $M$. Let $b_{original}$ be the number of bits used to represent each token in the original, uncompressed fixed-length format (e.g., 2×kmer length for DNA k-mers). The Compression Ratio is then:

$$CR(s, M) = \frac{\sum_{k=1}^{N} L_M(s^k)}{N \cdot b_{original}} \tag{3}$$

The numerator represents the total compressed size of the sequence $s$ in bits when encoded using model $M$. The denominator represents the total original size of the sequence in bits. A lower $CR$ indicates better compression, reflecting higher statistical regularity in the sequence $s$ as captured by the model $M$. This normalized metric facilitates comparisons of compressibility across different sequences and data domains.

When comparing two different compression models, $M_l$ and $M_m$, fit on distinct data sources (e.g., source $l$ and source $m$), we can assess their relative effectiveness on a specific sequence $s$. If $s$ originates from the same source distribution as the training data for $M_m$, the difference:

$$\Delta CR = CR(s, M_l) - CR(s, M_m) \tag{4}$$

quantifies the loss in compression efficacy when using a model trained on a different data source $M_l$ compared to the model trained on the native source $M_m$. A positive $\Delta CR$ indicates that model $M_l$ is less efficient for sequence $s$, suggesting a quantifiable difference between the underlying statistical distributions of the data sources $l$ and $m$.

---

**Algorithm 1** Huffman Coding Pseudocode

---

**Require:** Symbols $X = \{x_1, \ldots, x_n\}$ with frequencies $F = \{f_1, \ldots, f_n\}$
 1: **function** HUFFMAN$(X, F)$
 2:      $Q \leftarrow$ min-heap containing one leaf $(x_k, f_k)$ for each $k$
 3:      **while** $|Q| > 1$ **do**
 4:          $(a, f_a) \leftarrow Q.\text{extractMin}()$
 5:          $(b, f_b) \leftarrow Q.\text{extractMin}()$
 6:          $Q.\text{insert}(\text{Node}(a, b, f_a + f_b))$
 7:      **end while**
 8:      root $\leftarrow Q.\text{extractMin}()$
 9:      codes $\leftarrow$ `dict{}`
10:      ASSIGN(root, `""`)
11:      **return** codes
12: **end function**
13:
14: **function** ASSIGN(node, prefix)
15:      **if** ISLEAF(node) **then**
16:          codes[node.symbol] $\leftarrow$ prefix
17:      **else**
18:          ASSIGN(node.left, prefix $+$ 0)
19:          ASSIGN(node.right, prefix $+$ 1)
20:      **end if**
21: **end function**

---

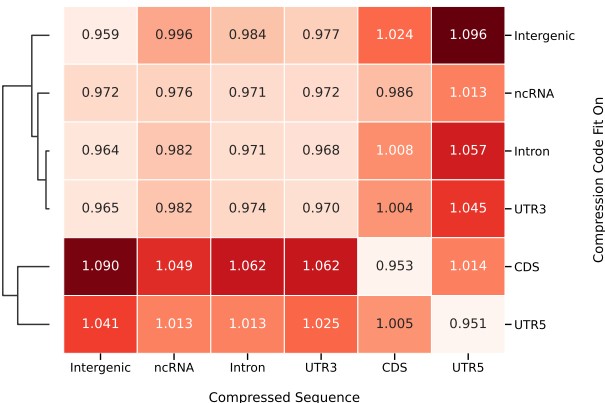

Figure 7: Cross-compression test scores by compression source.

For the compression algorithm we utilize Huffman coding, shown in Algorithm 1, which is a widely used algorithm for lossless data compression. Its strategy is to assign shorter binary codes to more frequent symbols and longer codes to less frequent ones, thereby minimizing the average code length for a given text or data sample.

Specifically, Huffman compression relies on empirically estimating the probability of discrete events from a given data sample. Based on these estimated probabilities, the algorithm assigns variable length binary codes to each event. The length of these codes is inversely proportional to the symbol's probability: more frequent symbols receive shorter codes. For an event $x_i$ with probability $p(x_i)$, the optimal code length is $-\log_2 p(x_i)$ bits. Huffman coding aims to construct a prefix code that approaches this theoretical optimum in terms of average code length.

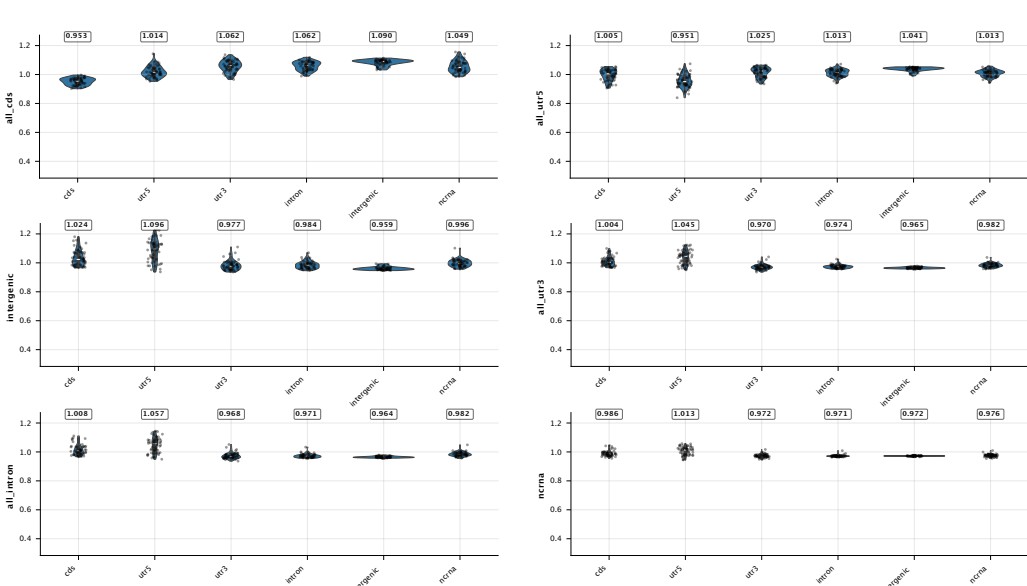

Figure 8: The raw distributions of 50 sequence data points compressed across Huffman codes fit to different reference sequences. 5' UTR values demonstrate higher variability than the rest.

Let $P_M$ be the probability distribution of tokens (symbols) implicitly learned by a compression model $M$ during its training (e.g., from the frequencies of k-mers in a training sequence). The average code length assigned by model $M$ to tokens from this distribution $P_M$, denoted $L_M$, satisfies $H(P_M) \le L_M < H(P_M) + 1$, where $H(P_M)$ is the Shannon entropy of the source distribution $P_M$:

$$H(P_M) = -\sum_{x \in X} P_M(x) \log_2 P_M(x) \tag{5}$$

Here, $X$ is the set of all possible tokens. This indicates that Huffman coding is nearly optimal for the data distribution it was trained on.

Similarly how compression ration $CR$ of a sequence $s$ has a close connection to the underlying entropy of a distribution, we can draw a similar connection to measuring cross entropy between two distributions. This is done by applying a compression model $M$ fit on $P_m$ to compress a sequence sampled $s_T$ from an underlying probability distribution $P_T$.

Consider two probability distributions defined over the same sample space $P_M$ and $P_T$. Assume sequence $s_T$ is drawn from $P_T$. When we use a compression model M (which was fit on $P_M$) to compress $s_T$, the average number of bits per token used to encode $s_T$ is

$$\frac{1}{N} \sum_{k=1}^{N} L_M(s_T^k). \tag{6}$$

This empirical average code length is an estimate of the cross-entropy between the true distribution $P_T$ and the model's distribution $P_M$:

$$H(P_T, P_M) = -\sum_{x \in X} P_T(x) \log_2 P_M(x) \tag{7}$$

The term $L_M(s^k)$ represents the code length assigned by model $M$ to the specific token $s^k$. In the case of using Huffman coding compression $P_M(x)$ are the probabilities used to construct the Huffman codes in model $M$, then $L_M(x) \approx -\log_2 P_M(x)$ for each token $x$.

The numerator of the Compression Ratio $CR$ 3 is the total compressed size of sequence s using model M. Thus, $CR(s_T, M) \cdot b_{original} = \frac{1}{N} \sum_{k=1}^{N} L_M(s^k)$, which is the empirical average bits per token. Therefore, the Compression Ratio $CR(s_T, M)$ is directly proportional to this empirical estimate of $H(P_T, P_M)$.

It is important to note that this analysis, based on token frequencies and Huffman coding, has inherent limitations. The choice of token size (e.g., k-mer length) is critical: larger tokens can capture more local context but may result in an exponentially larger sample space. This can make it challenging to empirically estimate the true probability distributions from finite training data, potentially leading to inaccurate frequency counts for rare events. In addition certain structures within the genome have a specific repetition order such as an open reading frame in coding sequences consisting of a sequence of codons composed of three nucleotides each. We use k-mer length of six to capture this regularity.

Furthermore, by focusing on individual token frequencies, this approach largely ignores global sequence structure and long-range dependencies. Such larger-scale structures, including regulatory motifs, codon usage patterns beyond local k-mers, and repetitive elements, are known to be significant features in both coding and intergenic regions and are not directly captured by this type of local, frequency-based compression analysis.

Training data for the compression analysis was generated using Genome Karyotype (genome-kit) (DeepGenomics, 2025) and GENCODE Basic version 47 annotations (Mudge et al., 2024). From these annotations, we isolated 3' UTRs, 5' UTRs, introns, and coding sequences (CDS) of protein-coding genes.

Because intergenic regions are vast, we restricted our intergenic sequence set to those on chromosome 1, which is about 10% of the human genome. This selection helped balance the corresponding length of different genomic region types in our training data, reducing potential biases in the compression model fitting that could arise (Table 7). For genes with multiple transcripts, we selected the canonical isoform using the APPRIS database (Rodriguez et al., 2013) to ensure consistency and reduce redundancy from alternative splice variants.

Table 7: Distribution and characteristics of sequences with and without repeats across various genomic regions. Counts represent the number of distinct sequences. Lengths are reported in base pairs (bp). The number of bases used for estimating the compression is omitting repeats and (*) indicates that we use fewer bases for computational efficiency.

| Genomic Region | Num. Sequences with Repeats | Total Length with Repeats (bp) | Num. Sequences without Repeats | Total Length without Repeats (bp) |
|---|---|---|---|---|
| CDS | 3,706 | 4,927,801 | 3,121 | 3,883,277 |
| Intron | 2,922 | 91,533,879 | 415 | 569,014 |
| UTR3 | 3,506 | 5,271,453 | 2,241 | 1,902,431 |
| UTR5 | 3,499 | 629,402 | 3,127 | 496,448 |
| Intergenic(chr1) | 196 | 13,640,727 | 4,157 | 148,795,829* |
| ncRNA | 38,016 | 33,615,313 | 14,648 | 9,549,965 |

When preparing training data for each category, we excluded a hold-out set of 50 unique sequences of variable lengths. These sequences were never seen by the compression algorithm during its fitting stage and were used as a test set. For the Huffman compression algorithm, all sequences (training and test) were first divided into non-overlapping k-mers of length six. Sequences were then truncated at their 3' end to ensure their total length was a multiple of six, losing at most five nucleotides per sequence. This preprocessing was crucial to keep the reading frame intact when sequences within each category were combined for frequency analysis by the compression algorithm.

Finally, to check if repetitive elements affected our findings, we conducted a parallel analysis where repeat regions, identified by RepeatMasker (Bao et al., 2015), were excluded from all genomic categories. The results from this repeat-masked analysis were very similar to those in the main paper, suggesting common repeats did not significantly affect our conclusions (Figure 9).

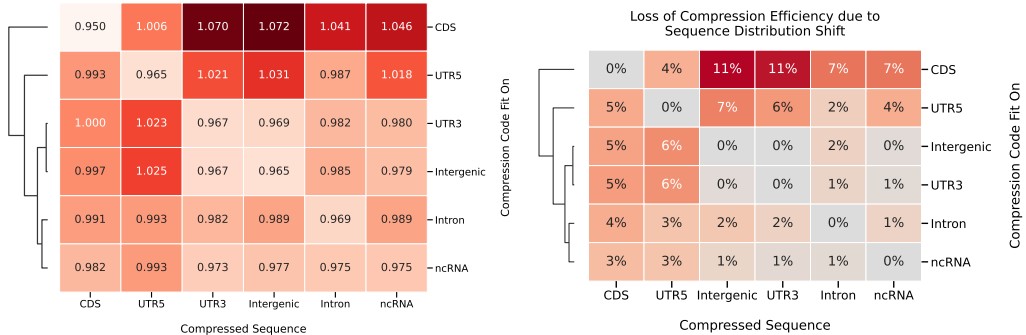

Figure 9: **Left:** Compression ratios indicating the percentage of size reduction associated with applying Huffman coding algorithm on the source data with k-mer length set to 6. All regions that contain repeats as indicated by repeat masker are omitted. **Right:** Cross-compression test scores by compression source. Numbers show percent increase in compression ratio while omitting genomic regions containing repeats.

