# OpenReview forum: "mRNABench: A curated benchmark for mature mRNA property and function prediction"
_ICLR.cc/2026/Workshop/LMRL — ICLR 2026 Workshop LMRL Poster_

### Official Review · Reviewer_DEtx · 2026-02-20
**Extensive and insightful benchmark of language models on mRNA sequences**

**Rating:** 7
**Confidence:** 4

**Review:**

## Overview:
This paper introduces mRNAbench, the first benchmark of sequence language models specifically focusing on messenger RNAs. The benchmark spans 79 property prediction tasks, 11 datasets and 60 sequence foundation models. The code is made available. In addition, the authors propose to improve one of the benchmarked models by modifying its pretraining objective. Using the modified model, they reach the best overall performance of the benchmark.

## General appreciation:
This paper addresses a critical gap in the literature since no benchmark has been specifically proposed for messenger RNAs, whereas their distinctive characteristics with respect to ncRNAs (especially reliance on sequence and codon organization rather than structure to perform their function) justify the establishment of a specific benchmark. The benchmark is comprehensive (60 models, 11 datasets, 79 benchmark tasks), and the code is made available, which will be useful to the community. The design of the benchmark is biologically well motivated, and the results are analyzed with interesting biological insights. The writing is very clear . However, the methodological innovation proposed is low. Therefore, I strongly recommend accepting the paper, but it should be regarded as a benchmark rather than a methodological contribution.

## Pros and cons
**Pros:**
* A benchmark of sequence foundation models specific to mRNAs is both a novel and biologically relevant contribution
* The benchmark is comprehensive (60 foundation models, 79 tasks, 11 datasets)
* The code is made available
* The splitting strategy, which is a crucial point in RNA biology but often overlooked in methodological or benchmarking papers, is extensively discussed
* Throughout the paper, the authors demonstrate a good understanding of the biological context of the benchmarked models
* The writing is very clear and pedagogical

**Cons:**
* The methodological innovation proposed is very incremental, since it consists of retraining Orthrus model by adding a masked language modeling pretraining objective to its initial contrastive pretraining objective
* Typo: MLR-HL-pair instead of MRL-HL-pair

---

### Official Review · Reviewer_TqRE · 2026-02-24
**Recommend accept, the authors have presented a novel dataset which addresses a current limitation in the field of mRNA representation. The authors show improved performance using their dataset despite training on substantially fewer parameters.**

**Rating:** 7
**Confidence:** 3

**Review:**

This paper presents the a novel dataset of annotated mRNA sequences, curated from multiple existing datasets. The authors trained and benchmarked a broad set of architectures and show that training on mRNA directly improves performance. The wide set of tasks and dataset size make mRNABench a useful contribution to the community.

In summary the authors have presented a novel dataset which addresses a current limitation in the field of mRNA representation. The authors show improved performance using their dataset despite training on substantially fewer parameters.

Comments:
* The authors could reduce Table 1 for clarity and sort by method type. This would aid interpretation. A bar graph could also be illustrative. It is also not clear which benchmarks were retrained or only evaluated with new heads on the mRNA set.
* The authors could also explicitly state in the main manuscript how many individual  mRNA samples are available per task and subtask.
* The authors could provide more supporting references in section 3 of the introduction.

---

### Official Review · Reviewer_8ar1 · 2026-02-25
**This paper, mRNABench shows that:  1. Scaling alone isn’t enough—biological priors and objective design are critical.  2. mRNA-specific benchmarks are essential since DNA/ncRNA-trained models don’t transfer well.  3. Proper evaluation requires homology-aware splits to avoid inflated performance.**

**Rating:** 7
**Confidence:** 3

**Review:**

Contributions:

1. Python package: lightweight access to datasets, tasks, and model wrappers.
2. Standardized evaluation: enabling fair comparison across architectures and training strategies (biologically informed data splitting strategies considered).
3. Standardized evaluation across 60 models and 259K experiments, each conducted with 10 random seeds, with statistically significant results.
4. Fairly measured representational quality without confounding factors by using linear probing.
5. Combining masked language modeling (MLM) with contrastive learning (CL) improves coverage across global and local tasks. Orthrus+MLM beats or matches SOTA in 7 of 11 datasets.

Significance:
The paper is significant because it provides the first rigorous, standardized, and biologically grounded benchmark for mRNA representation learning. It not only enables fair comparison across models but also uncovers principles for designing efficient, biologically aware objectives and thus accelerating progress in computational biology and genomics.

Cons:

1. Transcript-level embeddings are computed by averaging nucleotide embeddings. Doesn't this discard positional and structural information?

2.  When a model’s context length is too short to handle full transcripts, mRNABench chunks sequences and then averages embeddings. This is a practical solution, but it risks losing dependencies that span across chunk boundaries. The authors do note that overlapping chunks had “no major effect on performance." Is it because many of these tasks are evaluated via linear probes on averaged embeddings? Averaging tends to smooth out positional information, so overlap doesn’t add much signal in this setup.

3. While the paper compares against multiple baseline models, the evaluation would be strengthened by including more recent state-of-the-art methods, such as mRNA2vec for TE tasks and other recent models for the remaining tasks (if available).

---

### Meta-Review · Area_Chair_NXVj · 2026-02-28

**Recommendation:** Accept (Poster)
**Confidence:** 4

**Metareview:**

Accept

---

### Decision · Program_Chairs · 2026-03-02

**Decision:**

Accept (Poster)

**Comment:**

Please see the meta-review.